# The Role of Adipose Tissue and Nutrition in the Regulation of Adiponectin

**DOI:** 10.3390/nu16152436

**Published:** 2024-07-26

**Authors:** Sara Baldelli, Gilda Aiello, Eliana Mansilla Di Martino, Diego Campaci, Fares M. S. Muthanna, Mauro Lombardo

**Affiliations:** 1Department for the Promotion of Human Science and Quality of Life, San Raffaele Open University, Via di Val Cannuta, 247, 00166 Rome, Italyelianaayelen.mans@studenti.uniroma5.it (E.M.D.M.);; 2IRCCS San Raffaele Roma, 00166 Rome, Italy; 3Pharmacy Department, Faculty of Medicine and Health Sciences, University of Science and Technology-Aden, Alshaab Street, Enmaa City 22003, Yemen

**Keywords:** adipose tissue, adiponectin, metabolic homeostasis, insulin sensitivity, dietary interventions, metabolic diseases, type 2 diabetes, cardiovascular disease, fatty acid metabolism, endocrine function, adipokines, nutritional therapy, physical activity

## Abstract

Adipose tissue (AT), composed mainly of adipocytes, plays a critical role in lipid control, metabolism, and energy storage. Once considered metabolically inert, AT is now recognized as a dynamic endocrine organ that regulates food intake, energy homeostasis, insulin sensitivity, thermoregulation, and immune responses. This review examines the multifaceted role of adiponectin, a predominant adipokine released by AT, in glucose and fatty acid metabolism. We explore the regulatory mechanisms of adiponectin, its physiological effects and its potential as a therapeutic target for metabolic diseases such as type 2 diabetes, cardiovascular disease and fatty liver disease. Furthermore, we analyze the impact of various dietary patterns, specific nutrients, and physical activities on adiponectin levels, highlighting strategies to improve metabolic health. Our comprehensive review provides insights into the critical functions of adiponectin and its importance in maintaining systemic metabolic homeostasis.

## 1. Introduction

Adipose tissue (AT), primarily composed of adipocytes specializing in lipid control and metabolism, serves as the body’s principal energy reservoir. Previously considered a metabolically inert tissue, recent research has revealed its extensive regulatory influence on various physiological processes throughout the body. These functions encompass the regulation of food intake, maintenance of energy homeostasis, modulation of insulin sensitivity, thermoregulation, and immune responses [1]. It is considered a metabolically active endocrine organ, which produces specific hormones called adipokines. In addition to these hormones, adipose tissue generates genetic material, lipids, and proteins essential for its functioning. Furthermore, AT appears to play a primary role in glucose homeostasis together with liver and skeletal muscle by responding to stimulation by circulating lipids, insulin, hormones, and catecholamines [2]. 

AT’s fundamental function is the regulation of insulin and energy levels throughout the body. By secreting adipokines (e.g., adiponectin), AT regulates the levels of insulin and lipids that would otherwise collect in other tissues, contributing to insulin resistance, and thus favoring prediabetes. The cessation of any of the above functions contributes to the development of obesity-related metabolic diseases, such as type 2 diabetes, cardiovascular disease, fatty liver disease, and at least 13 types of cancer [3]. 

Adiponectin, a protein hormone predominantly released by white adipose tissue, has a crucial function in controlling glucose and fatty acid metabolism. It is the predominant adipokine found in the bloodstream, and its levels in circulation are negatively associated with insulin resistance, a key characteristic of various metabolic diseases [4].

In this review, we explore the multiple roles of adiponectin and other adipokines produced by adipose tissue. We discuss the regulatory mechanisms of adiponectin, its physiological effects, and its potential as a therapeutic target for metabolic diseases. Furthermore, we examine the impact of different dietary patterns, specific nutrients and physical activities on adiponectin levels, highlighting potential strategies to improve metabolic health. Through this comprehensive review, we seek to provide insights into the critical functions of adiponectin and its importance in maintaining systemic metabolic homeostasis.

## 2. Adipose Tissue

### 2.1. Adipose Tissue Growth

It is the main depot of triglycerides (TG) in mammals, made up of adipocytes: cells specialized in storing the primary energy reserve of the organism and responsible for synthesizing endogenous TG and releasing them as glycerol and fatty acids. Adipocytes originate from a complex process regulated by different hormones and intracellular pathways called adipogenesis [5]. Adipogenesis is a mechanism of proliferation and differentiation that involves the transformation of mesenchymal stromal/stem cells (MSCs) into lipid-containing mature adipocytes [6]. It is a process that occurs during childhood and adolescence and determines how many adipocytes will make up the adipose tissue in adults. Furthermore, this process defines the amount of lipids that can be retained in adipose tissue and, consequently, the fat mass in adults. Therefore, understanding the molecular mechanisms and potential targets in this cellular process would allow for us to modulate adipogenesis and, at the same time, obesity.

Adipogenesis is a process governed by many molecular factors and cellular activities capable of organizing the transformation of MSCs into preadipocytes and finally into mature adipocytes. It is essentially based on two phases: commitment and terminal differentiation [7]. During the commitment phase, MSCs are converted into preadipocytes, thus losing the ability to differentiate into osteocytes, chondrocytes, myocytes, etc. Then, different types of transcription factors, molecules, and genes trigger various signaling processes that lead preadipocytes to transform into mature adipocytes during the terminal differentiation phase [5]. More specifically, during the commitment process, the pluripotent stem cells positioned in the adipose tissue capture factors and signals that allow for them to begin their differentiation into preadipocytes. Subsequently, these cells are able to continue the differentiation process until they become adipocytes. Obviously, during this process, different groups of transcription factors come into play. There are many transcription factors that have been identified by researchers over the years, but in this review, we will focus on those that are activated in fibroblasts and preadipocytes [8]. These include the CCAAT/enhancer binding protein (C/EBP) gene family, peroxisome proliferator-activated receptor-γ (PPARγ), Krüppel-like factor (KLF) receptor, and sterol regulatory element binding protein (SREBP) gene family [9]. They are considered the main regulators that direct adipogenesis in a controlled manner. In particular, PPARg is essential for the differentiation of preadipocytes; C/EBPalpha is instead essential for adipocytes to become sensitive to insulin. Growth factors (TGF-beta, IGF-I) and hormones (insulin) also direct the terminal differentiation phase into adipocytes [10]. Angiogenesis and vascularization also contribute to the development of AT, in which the main protagonist is vascular endothelial growth factor (VEGF). Angiogenesis and vascularization are processes necessary for oxygenation and the transport of nutrients and hormones in the AT [11]. Dysfunctions of these two processes lead to AT dysfunction accompanied by inflammation and apoptosis. Two other factors that play a fundamental role in the vascularization of the AT are anti-angiogenic transcription factor forkhead box O1 (FOXO1) and the angiogenic adipokine neuregulin 4 (NRG4). It was indeed shown that, after a high-fat diet, FOXO1-deficient mice showed a significant accumulation of adipose tissue, an increase in glucose tolerance, and a decrease in fasting blood sugar [12]. The scientific literature considers NRG4 to be a pro-angiogenic adipokine. Expressing it constitutively in adipocytes leads to a decrease in hypoxia in the AT of obese mice, increasing the formation of adipose blood vessels and improving glucose tolerance [13].

From a historical point of view, AT is divided into white adipose tissue (WAT) and brown adipose tissue (BAT), so called due to their color. These types of tissues are well organized in defined areas of the body and present well specialized physiological and tissue differences. WAT is critically important for energy storage, endocrine communication and insulin sensitivity [14]. WAT secretes many biologically active substances and proteins with versatile functions. The latter are peptides that act as hormones or messengers regulating metabolism, including leptin, a hormone that is crucial for energy balance and is mainly produced by white adipose tissue, giving it an endocrine function [15]. WAT constitutes the largest volume of adipose tissue in most mammals. Its cells contain a single spherical lipid drop, delimiting the other organelles at the cell periphery. WAT is classified based on its anatomical location as subcutaneous or visceral. The visceral fat is located in the peritoneal cavity while the subcutaneous fat is positioned under the skin in the abdominal and gluteal–femoral deposits [16].

BAT uses energy to produce heat, a fundamental factor for maintaining body temperature. Recent studies have shown that BAT is not only present in newborns, but also in specific areas of adults (supraclavicular and thoracic), playing fundamental physiological roles: oxidation of body fat and diet-induced thermogenesis [17]. The cells that make up BAT have small and numerous lipid droplets together with a large number of iron-containing mitochondria. It is the uncoupled protein UCP1, contained in these mitochondria, that allows for BAT to carry out its thermogenic function. In fact, UCP1 disperses part of the proton gradient that is generated along the mitochondrial transport chain [18]. Adipocytes that make up BAT have small vacuoles containing TG. Furthermore, this tissue is richly vascularized due to a greater oxygen demand and is abundantly innervated compared to WAT [19].

AT also includes two additional types: beige and pink. Beige adipocytes are formed in the subcutaneous WAT by differentiating from white adipocytes or from a group of distinct preadipocytes. Beige adipocytes are of the thermogenic type, and many studies have shown that the browning of WAT can protect the individual (human, rodent) from obesity and metabolic dysfunction [20,21,22].

During pregnancy and breastfeeding, pink adipocytes are formed from breast WAT cells that transdifferentiate into milk-secreting glands rich in cytoplasmic lipids. In the post-lactation period, pink adipocytes transdifferentiate again into WAT and BAT [23].

Finally, a new type of adipose tissue has been identified in recent years: marrow adipose tissue (MAT) is located in the skeleton and contributes to local and systemic metabolic processes. The yellowish color of the bone marrow is in fact due to the presence of MAT inside it [24]. The adipocytes that make up the MAT derive mostly from the myogenic factor 5 positive progenitor (Myf5+) and contain a multilocular lipid droplet and UCP-1 activity [25]. They have the main function of transforming energy into heat (Figure 1). 

### 2.2. Metabolic Pathways in WAT

WAT uses two opposite metabolic pathways to satisfy the energy needs of various organs: lipogenesis and lipolysis, both driven by hormones, neuronal signals, and transcription factors [26]. This paragraph will analyze their mechanisms in detail.

After a meal, WAT is able to assimilate nutrients in the form of lipids through the bloodstream. This process called de novo lipogenesis (DNL), involves the absorption of fatty acids and the conversion of nutrients such as glucose into lipids [27]. The actor that directs this process is insulin, which by stimulating glucose absorption, promoting DNL, and suppressing lipolysis allows for the storage of lipids in adipocytes. Adipocytes use lipoprotein lipase (LPL) to store free fatty acids (FFA) retained by chylomicrons and very low-density lipoproteins (VLDL) [28]. LPL hydrolyzes TG, liberating fatty acids and monoacylglycerol. Once synthesized by the adipocytes, LPL is translocated to the vascular endothelium (luminal surface) and, to become fully functional, requires the presence of two additional proteins: apoA-V and GPIHBP1 (glycosylphosphatidylinositol-anchored high-density lipoprotein-binding protein 1) [29]. Once the FFAs are released, FATP1 (long-chain fatty acid transport protein 1) and CD36 facilitate their entry into the adipocytes, where they are activated by acyl-CoA synthetase to generate acyl-CoA [30,31]. Finally, the diacylglycerol acyltransferase enzymes (DGAT1 and DGAT2) combine an acyl-CoA and a diacylglycerol to form TG for storage in lipid droplets [32]. Taking a step back, this mechanism originates from the tricarboxylic acid (TCA) cycle: citric acid leaves the mitochondrion, and in the cytoplasm, it is converted into acetyl-CoA, malonyl-CoA, and finally into fatty acids. Both the production of malonyl-CoA (which induces a block in the oxidation of fatty acids), and the induction of particular enzymes regulate the DNL process at different levels. Among these factors, carbohydrate response element-binding protein (ChREBP), liver X receptor alpha (LXRa), and sterol response element-binding protein 1c (SREBP1c) play indispensable roles in activating key enzymes in the DNL process, such as fatty acid synthase (FAS) and acetyl-CoA carboxylase (ACC) [33,34,35,36].

From a biochemical point of view, the hydrolysis of triacylglycerols into FFA and glycerol occurs in a process called lipolysis [37]. As previously mentioned, the fatty acids present in WAT constitute a large energy reserve; therefore, when energy demands rise compared to normal physiological conditions (physical exercise, fasting), adipocytes make their triacylglycerols available as a source of energy for peripheral tissues. 

### 2.3. Metabolic Pathways in BAT

Using positron emission tomography (PET) with the glucose analog tracer 18F-fluorodeoxyglucose (18FDG), it was possible, in 2003, to identify BAT for the first time [38,39]. To date, the thermogenic characteristic of this tissue represents an important objective for the treatment of obesity and heart diseases linked to metabolic problems. In fact, it is able to produce heat through uncoupled oxidative phosphorylation. There are two types of BAT-driven thermogenesis: (i) cold-induced thermogenesis and (ii) diet-induced thermogenesis [40,41]. Regarding exposure to cold, the efferent fibers of the sympathetic nervous system release norepinephrine, which activates the lipolysis of TG, releasing fatty acids that in turn activate pro-thermogenic genes, such as UCP1 [42,43]. Other non-sympathetic molecules such as triiodothyronine (T3), hepatic bile acids, and various retinoids have also been shown to play a role in the thermogenic activation of brown adipocytes [44]. The protagonist of diet-induced thermogenesis is UCP1, which is present at high levels in BAT adipocytes compared to ATP synthase, an enzymatic complex necessary to produce ATP. As a consequence, a reduction in the proton gradient occurs, and energy/heat is dissipated throughout the body through the widespread vascularization of the BAT [45]. The coactivator protein PPARg coactivator-1a (PGC1-α) also comes into play during exposure to cold, regulating mitochondrial biogenesis [46,47]. p38 mitogen-activated protein kinase (MAPK) phosphorylates PGC-1α in response to norepinephrine [48]. PGC-1α has the main role of coactivating PPAR, ESRR, the thyroid receptor, and interferon regulatory factor 4 (IRF4). In this way, an increase in the activity of UCP1 and other genes involved in thermogenesis and mitochondrial biogenesis increases. Along with PGC-1α, the mTOR (mammalian target of rapamycin) pathway is also stimulated by norepinephrine. mTOR is a serine threonine protein kinase that plays an important role in the function and development of adipocytes through the activity of its two complexes, mTORC1 and mTORC2. In detail, mTORC1 and mTORC2 regulate thermogenesis by modulating lipolysis and lipogenesis, mitochondrial biogenesis and functionality, and the expression of thermogenic genes. Furthermore, mTORC2 promotes glycolysis, thereby regulating glucose homeostasis. In detail, some authors demonstrate that mTORC2 is able to regulate insulin-induced glucose absorption and is activated in BAT through B-adrenergic stimulation. Furthermore, mice that lack mTORC2 at the adipose tissue level present reduced glucose absorption and glycolysis [49]. Other authors show that WAT mTORC2 is an important player of a “nutrient sensing” mechanism capable of controlling glucose homeostasis/flow by modulating cAMP Response Element-Binding Protein (CREB) activity and de novo lipogenesis [50].

### 2.4. Inflammation in AT

In recent years, WAT has always been defined as an endocrine organ capable of regulating inflammation and angiogenesis through the production of cytokines and adipokines. An increase in caloric intake linked to an increase in weight/obesity determines a transformation of the WAT: its adipocytes become inflamed and dysfunctional, and it also undergoes the infiltration of cells of the immune system [51]. These cells together with the dysfunctional adipocytes begin to secrete proinflammatory cytokines that make the AT and nearby organs dysfunctional. Among these, interleukin-6 (IL-6) and tumor necrosis factor-a (TNF-α) are those most produced by WAT, especially in obese subjects [52]. Other cytokines/chemokines also participate in the inflammatory response in the WAT of obese subjects or those with type 2 diabetes: IL-1, IL-8, Monocyte Chemoattractant Protein-1 (MCP-1), and macrophage inflammatory protein 1 [53,54]. Upstream, the production of these cytokines/chemokines is governed by two signaling pathways: c-Jun N-terminal kinases (JNK) and nuclear factor kappa-light-chain-enhancer of activated B cells (NFkB). In particular, JNK activates the transcription of proinflammatory genes and, by phosphorylating the IRS-1 (Insulin Receptor Substrate-1) receptor, inhibits the insulin signaling pathway [55]. NFkB, on the other hand, induces the gene expression of TNFα, IL-6, and MCP1.

Recently, the importance of another family of cytokines has emerged: IL-1 (IL-1β and IL-18), whose production is controlled by the Nod-Like Receptor (NLR) family pyrin domain-containing 3 (NLRP3) inflammasome [56]. IL-1β and IL-18 are abundantly produced during obesity and aging [57]. Specifically, following signaling by LPS and lipids (both abundant in obese subjects), TLR4, IL-1R, and TNF receptor-associated factor 2 (TRAF2) are activated. NFkB subsequently enters the nucleus mediates transcription of NLRP3 and pro-IL-1β, which allows for subsequent assembly of the NLFPR3 inflammasome [58]. Once assembled, procaspase-1 is cleaved into active caspase-1, which mediates the transformation of gasdermin D, pro-IL-1β, and pro-IL-18 into their biologically active forms. As mentioned above, being a hormone with anti-inflammatory effects, adiponectin regulates the activity of inflammation [59]. Its agonist (AdipoRon) blocks the activity of the inflammasome through the 5′ adenosine monophosphate-activated protein kinase (AMPK), FoxO4, and NFkB pathway [60]. In addition to adiponectin, leptin also influences the activity of the inflammasome. It has in fact been demonstrated that leptin is able to promote the production of IL-18 mediated by NLRP3 contributing to the production of reactive oxygen species (ROS) [61].

However, little is known about the activation of the inflammatory process at the BAT level in the case of metabolic dysfunction. BAT also has cells of the immune system, such as macrophages, neutrophils, and lymphocytes [62]. Inflammation that occurs in the BAT can contribute to its dysfunction, obesity, and associated metabolic problems. BAT can whiten due to inflammation. In detail, a diet rich in fat causes the transformation of a brown adipocyte into a unilocular cell similar to a white adipocyte. This is due to a mixture of factors including the infiltration of macrophages, the degeneration of brown adipocytes, and the formation of the crown structure (CLS). This type of bleached BAT exhibits significant activation of the NLRP3 inflammasome [63]. Thus, the proinflammatory cytokines produced influence the thermogenesis of BAT and its browning capacity, damaging the mechanism of glucose absorption and energy expenditure [64].

## 3. Adiponectin 

A hormone secreted by the adipose tissue (white adipocyte, WAT), adiponectin positively modulates the endocrine system and increases insulin sensitivity both in healthy subjects and in glucose and lipid metabolism disorders. WAT functions not only as an energy storage depot but also as an endocrine organ, releasing adiponectin, which enhances insulin sensitivity and exerts anti-inflammatory effects [65]. Additionally, brown adipose tissue (BAT) contributes to thermogenesis and energy expenditure, indirectly supporting adiponectin’s metabolic regulatory roles [66]. Furthermore, adiponectin promotes the browning of white adipocytes into metabolically active beige adipocytes, enhancing overall energy expenditure and providing protective effects against obesity and metabolic disorders [67]. Thus, adiponectin’s secretion and function are intricately linked with the physiological roles of WAT, BAT, and beige adipocytes, highlighting its significance in maintaining metabolic homeostasis. In contrast to other adipokines, blood concentration of adiponectin is reduced in obese individuals due to low physical activity and sedentary lifestyles. Adiponectin is present at high concentrations in plasma (3–30 μg/mL), which accounts for up to 0.05% of total serum protein. Low levels of adiponectin in the blood are negatively correlated with various types of neoplasia, cardiovascular disease, and type 2 diabetes mellitus. Moreover, several studies have shown its significant improvement due to a healthy diet. In addition, adiponectin shows a protective effect on neurons and neural stem cells and stimulates fatty acid oxidation in skeletal muscle by reducing TG accumulation [68]. Gender also influences adiponectin levels; indeed, the hormone has higher levels in women than in men, probably due to higher levels of estrogen, known to have an impact on adipose tissue. Understanding the structure of adiponectin and its interaction with specific receptors is crucial for developing therapeutic strategies targeting metabolic diseases.

### 3.1. Structure and Receptors

Adiponectin is encoded by the Adiponectin, C1Q And Collagen Domain Containing (ADIPOQ) gene in humans. The transcription of this gene is influenced by various nutritional and hormonal factors. For instance, insulin and glucocorticoids tend to suppress adiponectin gene expression. Following mRNA transcription, adiponectin undergoes several post-translational modifications—hydroxylation and glycosylation enhance its structural integrity and function. Adiponectin, secreted by the Golgi apparatus into the bloodstream, is distinguished by its oligomeric forms, which include low molecular weight (LMW) trimers, medium molecular weight (MMW) hexamers, and high molecular weight (HMW) multimers. The primary structure comprises 244 amino acids, featuring a signal sequence, a collagenous domain, and a globular domain. The formation of higher-order structures is facilitated by hydrophobic interactions and disulfide bonds, critical for its functional versatility. LMW trimers form the basic building block, which further associates into larger hexamers and multimers through non-covalent interactions. Each oligomer exhibits distinct physiological functions, suggesting a regulated mechanism of action dependent on the oligomeric state. The larger oligomers, especially HMW forms, have higher biological activity and are more effective in enhancing insulin sensitivity, exerting anti-inflammatory effects, and promoting fatty acid oxidation [69,70]. Adiponectin exists in both full-length and globular forms. The globular form, derived from the proteolytic cleavage of the full-length protein by thrombin, is constituted by the C-terminal globular domain and interacts with receptors to initiate specific signaling pathways that are distinct from those activated by the full-length form. Despite circulating in low levels in human plasma, the globular form may influence energy balance by promoting the oxidation of free fatty acids (FFA) in muscle mitochondria. The differential activities of these forms underpin their metabolic regulatory roles. Moreover, adiponectin has a relatively short half-life of 45 to 75 min, with minimal degradation occurring while it circulates. It is primarily cleared by the liver, although it can also bind to pancreatic beta cells, as well as certain cells in the heart and kidneys. Adiponectin mediates its effects primarily through two receptor isoforms, AdipoR1 and AdipoR2. AdipoR1 is ubiquitously expressed with high levels in skeletal muscle, whereas AdipoR2 is predominantly expressed in the liver where it enhances insulin sensitivity through the activation of peroxisome proliferator-activated receptor alpha (PPAR-α) [68]. These receptors are integral membrane proteins that facilitate adiponectin binding and subsequent activation of intracellular signaling cascades. AdipoR1 and AdipoR2 belong to a newly identified receptor family characterized by seven transmembrane domains, unlike classical G-protein-coupled receptors. This structural configuration suggests a unique mechanism of action, potentially involving ceramidase activity, which catalyzes the hydrolysis of ceramides to generate sphingosine and free fatty acids, playing a crucial role in mediating the metabolic effects of adiponectin [71]. The AdipoR1 receptor is a high-affinity receptor for the globular form and a low-affinity receptor for HMW adiponectin in skeletal muscle. In contrast, AdipoR2 has an intermediate affinity for the HMW form in the liver. Adiponectin binds its receptors to modulate energy expenditure, the inflammatory response, insulin sensitivity, and oxidative processes of energy substrates (especially lipids). The regulation of its synthesis is intricately linked to the body’s metabolic status, influenced by factors such as diet, fasting, and inflammation, making it a key target for potential therapeutic interventions in metabolic disorders and obesity. 

#### 3.1.1. Functions of Adiponectin in Different Organs of the Body

Although adiponectin is predominantly produced by adipocytes, it directly acts on the liver, skeletal muscle, and vasculature. Adiponectin exerts pleiotropic actions in various organs, i.e., it promotes insulin sensitivity, and apoptosis in cancer cells, primarily through its anti-inflammatory, anti-atherogenic, and insulin-sensitizing effects. Its widespread impact on different tissues makes it a crucial player in maintaining metabolic health serving as a potential target for therapeutic interventions in metabolic disorders. Figure 2 shows the beneficial effects of adiponectin across different organs, emphasizing key molecular pathways. In detail, specific pathways like the AMPK pathway are predominantly activated by AdipoR1 whereas the PPARα pathway is primarily activated by AdipoR2.

#### 3.1.2. Functions of Adiponectin in the Liver

In the liver, adiponectin inhibits gluconeogenesis, the process of glucose production from non-carbohydrate substrates, and enhances fatty acid oxidation. These actions improve insulin sensitivity and reduce hepatic lipid accumulation, thereby protecting against nonalcoholic fatty liver disease (NAFLD) and type 2 diabetes. The beneficial effects of adiponectin in the liver are mediated through the activation of AMP-activated protein kinase which inhibits the expression of key gluconeogenic enzymes, such as phosphoenolpyruvate carboxykinase (PEPCK) and glucose-6-phosphatase (G6Pase), thereby reducing hepatic glucose output [70,72]. Additionally, adiponectin upregulates peroxisome proliferator-activated receptor alpha (PPARα) pathways and its target genes, including carnitine palmitoyltransferase 1 (CPT1), which facilitates fatty acid transport into mitochondria for β-oxidation, reducing hepatic lipid accumulation and protecting against NAFLD and type 2 diabetes [73]. Moreover, adiponectin, by activating the liver kinase B1 (LKB1)/AMPK pathway, protects against liver injury by reducing oxidative stress and enhancing antioxidant defenses, by increasing the expression of enzymes like superoxide dismutase (SOD) and catalase, thus reducing oxidative stress and preventing apoptosis in liver cells [74]. Adiponectin exhibits anti-inflammatory properties by inhibiting the NFkB pathway, which decreases the production of proinflammatory cytokines such as TNF-α and upregulates anti-inflammatory cytokines like IL-10 [75]. It ameliorates liver injury caused by lipopolysaccharide (LPS) and other hepatotoxic agents by modulating inflammatory responses [76]. 

#### 3.1.3. Functions of Adiponectin in Kidneys

In the kidneys, adiponectin exerts protective effects by reducing oxidative stress and inflammation. It prevents the degradation of renal arteries, decreases protein excretion, and enhances renal filtration. These actions are crucial for maintaining kidney health and preventing chronic kidney diseases, especially in the context of metabolic syndrome [73]. Both AdipoR1 and R2 receptors are expressed in the kidneys. Adiponectin may protect the kidneys from albuminuria, exert antioxidant effects, and reduce inflammation through the activation of AMPK-related pathways, which leads to a reduction in NADPH oxidase activity, thereby decreasing oxidative stress and inflammation [77]. Furthermore, adiponectin enhances endothelial nitric oxide synthase (eNOS) expression through the AMPK pathway, improving endothelial function and reducing albuminuria [78]. Adiponectin reduces albuminuria and ameliorates glomerular hypertrophy and inflammation in diabetic nephropathy, indicating its role in protecting against kidney disease progression [78,79]. Analyzing adiponectin levels in non-diabetic hypertensive men revealed that lower adiponectin levels were associated with the presence of microalbuminuria [68]. Moreover, adiponectin also plays a protective role against renal ischemia–reperfusion injury by upregulating heme oxygenase-1 (HO-1) via the PPARα-dependent pathway, which is mediated through the enhancement of COX-2 and prostacyclin expression [80].

#### 3.1.4. Functions of Adiponectin in Skeletal Muscle

In skeletal muscle, adiponectin enhances glucose uptake and fatty acid oxidation, contributing to improved insulin sensitivity and energy consumption. These effects are primarily mediated by the activation of AMPK and p38 MAPK pathways. Upon adiponectin binding to its receptors, AdipoR1 and AdipoR2, AMPK is activated and promotes the translocation of glucose transporter 4 (GLUT4) to the cell membrane, enhancing glucose uptake [81]. Moreover, the activation of the AMPK pathway enhances mitochondrial function and reduces oxidative stress by upregulating antioxidant enzymes such as SOD and catalase. This activity mitigates oxidative damage and supports muscle cell health [82]. Additionally, adiponectin upregulates peroxisome PPARα and its coactivator PGC-1α, enhancing mitochondrial biogenesis and fatty acid oxidation in muscle cells [83]. This metabolic regulation in skeletal muscle is crucial for maintaining glucose homeostasis and preventing insulin resistance [65]. Moreover, adiponectin influences muscle fiber composition, promoting the maintenance of oxidative type I fibers and preventing the accumulation of lipid droplets within muscle cells [84]. Physical exercise increases circulating adiponectin levels and the expression of its receptors in skeletal muscle, which is associated with improved metabolic health and insulin sensitivity. 

#### 3.1.5. Functions of Adiponectin in Cardiovascular System

Adiponectin plays a protective role in the cardiovascular system by exerting anti-inflammatory and anti-atherogenic effects. It enhances nitric oxide production, reduces ROS in endothelial cells, and mitigates vascular inflammation. These actions improve endothelial function and reduce the risk of atherosclerosis and cardiovascular diseases. Additionally, adiponectin’s ability to attenuate smooth muscle cell proliferation and migration further supports its cardiovascular protective effects [85]. Moreover, adiponectin has beneficial effects on the heart through its interaction with APPL1 and AMPK by protecting the heart muscle through various mechanisms: it increases insulin-stimulated fatty acid and glucose uptake, as well as increasing AdipoR1 interactions with APPL1. The APPL1 protein binds to AMPKα2 causing phosphorylation of acetyl-CoA carboxylase (ACC), i.e., its subsequent inhibition, which leads to increased oxidative phosphorylation in cardiac tissue [68]. Moreover, adiponectin has beneficial effects on the heart by activating the PPARα pathway, which enhances mitochondrial biogenesis and fatty acid oxidation in cardiomyocytes. This activation reduces cardiac hypertrophy and fibrosis, contributing to the overall cardioprotective effects of adiponectin [86]. In addition, adiponectin modulates the phosphatidylinositol 3-kinase (PI3K) and Akt signaling pathways, which play crucial roles in protecting against cardiac apoptosis and improving cardiac function during ischemia–reperfusion injury [87]. In summary, adiponectin protects the cardiovascular system by activating multiple molecular pathways, including AMPK, APPL1, PPARα, and PI3K/Akt, which collectively enhance endothelial function, reduce inflammation, and improve cardiac metabolism and function.

#### 3.1.6. Functions of Adiponectin in the Central Nervous System

Adiponectin, once believed unable to cross the blood–brain barrier, controls various brain functions through peripheral circulation, influencing synaptic plasticity, energy homeostasis, and hippocampal neurogenesis. Adiponectin influences metabolic processes in the brain and has potential neuroprotective roles. Both AdipoR1 and AdipoR2 are expressed in different regions of the brain, including the hypothalamus, hippocampus, and cortex. The activation of these receptors mediates several signaling pathways including AMPK via AdipoR1 and PPARα pathways via AdipoR2 in the hypothalamus, which can regulate appetite, energy consumption and synaptic plasticity. Although its precise functions in the brain are still being elucidated, evidence suggests that adiponectin may play a role in neuroprotection and cognitive function [88]. Specifically, adiponectin promotes Janus kinase 2 (JAK2) and signal transducer and activator of transcription 3 (STAT3) phosphorylation, leading to enhanced neuronal survival and function. This pathway is particularly important for reducing apoptosis and promoting synaptic plasticity, which are essential for cognitive functions [89]. A reduction in neurogenesis at the latter site in adults may be linked to depression and chronic stress conditions. Conversely, adiponectin, following increases in its blood concentration, promotes hippocampal neurogenesis [68]. Adiponectin binding to its receptors, AdipoR1 and AdipoR2, leads to the recruitment and activation of the adaptor protein APPL1. This activation facilitates the cytosolic translocation and activation of LKB1, which subsequently activates AMP-activated protein kinase, which enhances the proliferation and differentiation of neural stem cells into neurons and glial cells. This process is crucial for maintaining neurogenesis, especially in the hippocampus, a brain region involved in learning and memory [90]. Moreover, through the activation of sirtuin 1 (SIRT1) and PPARγ, adiponectin inhibits the differentiation of Th17 cells and reduces autoimmune central nervous system inflammation [91].

#### 3.1.7. Functions of Adiponectin in the Bones

Adiponectin has been shown to influence bone formation and mineral density. It enhances the proliferation of osteoblasts, the cells responsible for bone formation, and regulates bone homeostasis. This function links adiponectin to both fat and bone density, suggesting that adiponectin could play a role in bone health and the prevention of osteoporosis [92]. Specifically, adiponectin increases bone mass by inhibiting osteoclast formation and activity while stimulating osteoblast differentiation and activity. Adiponectin regulates bone mass by decreasing FoxO1 activity in a PI3-kinase-dependent manner, which favors osteoblast survival and function [93]. Furthermore, adiponectin inhibits osteoclastogenesis by suppressing the NFkB and p38MAPK signaling pathways, reducing bone resorption [94]. Clinical studies show that serum adiponectin levels are negatively associated with bone mineral density (BMD), which measures the amount of minerals (such as calcium) in a specific volume of bone, indicating that higher adiponectin levels may be linked to lower bone mass [95].

### 3.2. Role in Diseases

Adiponectin participates in the modulation of lipid metabolism, energy regulation, inflammation, and insulin sensitivity. Adiponectin levels are inversely related to cancer, cardiovascular disease, and diabetes, and are influenced by nutritional factors, providing protective effects on neural stem cells and neurons. Table 1 outlines adiponectin’s role in various diseases, emphasizing its utility as a biomarker for these conditions and its potential as a target for monitoring the effectiveness of preventive and therapeutic interventions.

#### 3.2.1. Cardiovascular Disease

Low levels of adiponectin (hypoadiponectinemia) are associated with an increased risk of various cardiovascular diseases, including coronary artery disease, hypertension, and heart failure. Clinical studies have shown that increasing adiponectin levels through lifestyle modifications or pharmacological interventions can improve cardiovascular outcomes [107]. Adiponectin has significant anti-inflammatory properties, which help to mitigate the chronic inflammation associated with obesity and CVD. It inhibits the expression of proinflammatory cytokines and reduces the infiltration of inflammatory cells into the vascular wall. These effects are critical in preventing the progression of atherosclerosis [108]. Adiponectin enhances endothelial function by increasing nitric oxide production and reducing oxidative stress in vascular endothelial cells. This leads to improved vasodilation and reduced vascular inflammation, which are crucial for preventing atherosclerosis and maintaining vascular health [109]. Moreover, adiponectin also directly affects cardiac cells, offering protection against myocardial ischemia–reperfusion injury and pathological cardiac remodeling. It reduces cardiomyocyte apoptosis and fibrosis, thus preserving cardiac function and preventing heart failure [110]. 

#### 3.2.2. Type 2 Diabetes and Obesity

Adiponectin is crucial in improving insulin sensitivity and reducing inflammation. Its secretion decreases in obesity, which contributes to insulin resistance and the development of type 2 diabetes. Adiponectin enhances insulin sensitivity by stimulating glucose uptake in muscles and fatty acid oxidation in liver and muscle tissues [111]. Moreover, adiponectin has anti-inflammatory properties that mitigate the chronic low-grade inflammation seen in obesity and diabetes [112]. Several studies have identified genetic polymorphisms in the adiponectin gene that are associated with altered adiponectin levels and increased risk of type 2 diabetes. For instance, the T-G polymorphism in exon 2 of the adiponectin gene has been linked to increased body mass index (BMI) and decreased insulin sensitivity [113]. 

Lower plasma adiponectin levels have been consistently associated with insulin resistance. In rhesus monkeys, decreased adiponectin levels correlated with early stages of obesity and type 2 diabetes, highlighting its role in the progression of insulin resistance [114]. Similarly, higher adiponectin levels were associated with a lower risk of developing type 2 diabetes in diverse populations, demonstrating a dose–response relationship [115]. Long-established pharmaceutical therapies, such as metformin and thiazolidinediones, have been shown to increase the secretion of adiponectin and improve outcomes in patients with chronic diseases like type 2 diabetes. 

#### 3.2.3. Nonalcoholic Fatty Liver Disease (NAFLD) 

Nonalcoholic fatty liver disease (NAFLD) is a widespread liver disorder that ranges from simple steatosis (fat accumulation) to nonalcoholic steatohepatitis (NASH), potentially progressing to fibrosis, cirrhosis, and liver cancer. In patients with NAFLD, plasma adiponectin levels are significantly lower compared to healthy individuals. Typically, healthy individuals have adiponectin levels ranging from 3 to 30 µg/mL, whereas patients with NAFLD often show levels below 6 µg/mL. This reduction in adiponectin impairs its ability to activate AMPK and PPAR-α pathways, crucial for reducing hepatic glucose production and enhancing fatty acid oxidation [100]. Furthermore, the interaction between the decrease in total adiponectin levels and the different adiponectin isoforms is crucial in the context of NAFLD. Martínez-Huenchullán et al. (2023) demonstrated that both constant-moderate and high-intensity interval training (HIIT) affect adiponectin isoforms differently, which may impact cardiomyocyte metabolism. These findings are relevant for NAFLD, as the distribution of adiponectin isoforms, in particular the reduction in HMW adiponectin, is associated with liver disease severity. Lower levels of HMW adiponectin, which has potent anti-inflammatory and insulin-sensitizing effects, contribute significantly to metabolic dysfunction and liver inflammation in patients with NAFLD. This highlights the potential of targeted exercise interventions to modulate adiponectin isoforms and improve NAFLD symptoms [116].

#### 3.2.4. Alzheimer’s Disease

The relationship between insulin resistance and Alzheimer’s disease (AD) is well documented. Adiponectin enhances insulin signaling, which is crucial for maintaining neuronal health and function. In AD, dysregulated insulin signaling contributes to the disease’s neuropathology. Adiponectin’s ability to improve insulin sensitivity might thus play a protective role against AD [117]. In aged adiponectin knockout mice, a deficiency in adiponectin led to AD-like cognitive impairments and pathologies, including increased amyloid-beta (Aβ) deposition and tau hyperphosphorylation, which are characteristic of AD [118]. Moreover, genetic studies have identified specific adiponectin gene polymorphisms that are associated with an increased risk of late-onset AD. These polymorphisms may influence adiponectin levels and its receptor function, thereby affecting the progression of AD [119]. Interventions aimed at increasing adiponectin levels or mimicking its action could potentially alleviate some of the neuropathological features of AD. 

#### 3.2.5. Cancer

Numerous studies have highlighted the inverse relationship between adiponectin levels and cancer risk. For instance, lower adiponectin levels are associated with an increased risk of breast cancer, especially in postmenopausal women. In a study involving 174 women with newly diagnosed breast cancer and 167 controls, an inverse and significant association between serum adiponectin and breast cancer risk was observed by Mantzoros et al. [103]. Another study confirmed that women in the lowest tertile of serum adiponectin levels had a significantly higher risk of breast cancer compared to those in the highest tertile [104]. Adiponectin’s role extends to colorectal cancer as well. Research involving 18,225 men in the Health Professionals Follow-up Study found that men with lower plasma adiponectin levels had a significantly higher risk of developing colorectal cancer [105]. Additionally, a study in the European Prospective Investigation into Cancer and Nutrition showed that non-HMW adiponectin (non-HMW) was inversely associated with colorectal cancer risk, suggesting its crucial role in colorectal carcinogenesis [106]. These examples of evidence highlight the adiponectin’s potential role in cancer prevention and as a biomarker for early detection. 

## 4. Influence of Diet on Adiponectin

The concentration of adiponectin in the blood is influenced by numerous factors such as age, gender, body weight, percentage of adipose tissue, and, in particular, diet. Various dietary patterns and lifestyle interventions have been studied for their impact on adiponectin levels, shedding light on how specific diets and physical activity can improve metabolic outcomes [120,121,122,123]. We explore the influence of different dietary approaches, including low-energy diets, the Mediterranean diet, the DASH diet, vegetarian diets, and the ketogenic diet, on adiponectin levels, as well as the synergistic effects of diet and exercise. Energy restriction through low-calorie diets has been shown to effectively increase adiponectin levels, contributing to improved metabolic health. Studies have shown that weight loss achieved through calorie restriction and exercise can significantly increase adiponectin concentrations, with further benefits observed when dietary fiber is included. Similarly, adherence to the Mediterranean diet (MD), which emphasizes the consumption of fruit, vegetables, whole grains, and healthy fats, is associated with elevated adiponectin levels and improved lipid and glucose metabolism. 

### 4.1. Influence of Energy Restriction on Adiponectin Blood Levels

Low-energy diets have been shown to positively influence adiponectin concentrations. Lower body weight increases the blood concentration of this hormone, making the adoption of a negative energy balance advantageous as it effectively reduces body weight. Weight loss achieved through a low-calorie diet and exercise has been shown to increase adiponectin levels by 18–48%. Furthermore, fiber supplementation resulted in a 60–115% increase in adiponectin levels [124]. Salehi-Abargouei et al. conducted a systematic review and meta-analysis confirming the beneficial effects of low-calorie diets on adiponectin levels, providing solid evidence that caloric restriction effectively increases adiponectin, which is crucial for improving metabolic health [125].

Several studies have shown that the effect of calorie restriction on adiponectin levels is obtained independently of the calorie regime followed. A significant increase in adiponectin following weight loss was observed in a study involving 107 obese adult males randomized to follow a low-calorie regimen (reducing caloric intake by 500–700 kcal) compared to an isocaloric diet [126]. Similar results have been observed in other studies combining calorie reduction with physical activity in overweight subjects. For example, a weight-loss program involving a calorie reduction of 500–800 kcal/day combined with regular physical activity resulted in a significant increase in adiponectin levels among overweight and obese premenopausal women: participants experienced a 50.2 percent increase in adiponectin after 16 weeks of intervention [127]. In another study involving 143 overweight or obese adults who underwent a six-month behavioral intervention, weight loss was associated with increased levels of total adiponectin and HMW in the two study groups, both of which were on a fat-restricted hypocaloric diet, but one omnivorous and the other lacto-ovo-vegetarian [128]. 

In conclusion, low-energy diets have a significant impact on levels of adiponectin. Consistent results from various studies indicate that reducing body weight through calorie restriction, especially when combined with physical activity, leads to a substantial increase in adiponectin levels. These improvements in adiponectin are associated with improved metabolic health.

### 4.2. Mediterranean Diet

The MD promotes a high consumption of fruit, vegetables, fiber, unsaturated fats, whole grains, unrefined carbohydrates, a balanced consumption of dairy products and fish, and a reduced intake of saturated fatty acids and simple sugars [129]. Numerous studies (Table 2) have shown that high adherence to this nutritional pattern is associated with increased levels of adiponectin, a protein involved in the regulation of lipid and glucose metabolism as well as inflammatory responses.

An analysis of 20 randomized controlled trials confirmed that the MD increases adiponectin levels in the subjects studied [130]. Studies on type 1 diabetic patients found that a diet rich in unsaturated fats, such as those found in olive oil, increased adiponectin levels, while a diet rich in saturated fats reduced them [131]. MD components such as oleic acid and hydroxytyrosol, found in virgin olive oil, positively regulate adiponectin expression due to their anti-inflammatory and insulin-sensitizing properties [132]. Another study showed significant improvements in serum lipid profile and adiponectin levels with a low-calorie diet high in monounsaturated fats following the MD model [133]. The genomic mechanism that could explain this positive association involves the peroxisomal proliferator-activated receptor gamma (PPAR-γ) pathway. Polyunsaturated fatty acids such as eicosapentaenoic and docosahexaenoic stimulate adiponectin mRNA expression in adipocytes through activation of this pathway [125]. Protection against inflammation caused by metabolic syndrome and its comorbidities is enhanced by the consumption of low glycemic index foods, omega-3, and olive oil, all of which increase adiponectin levels in the blood [134]. 

A study of 133 obese patients carrying the 712 G/A rs3774261 polymorphism in the ADIPOQ gene found that non-carriers of the G allele showed significant improvements in serum adiponectin levels, lipid profiles, and C-reactive protein (CRP) in response to a low-calorie diet following an MD model [135]. Another study showed that 18 overweight/obese subjects fed a MD enriched with 40 g/day of high-quality extra virgin olive oil for three months had a significant decrease in proinflammatory cytokines and an increase in IL-10 and adiponectin [136]. In addition, a study of 11 overweight/obese men and women reported significant improvements in body fat, waist circumference, and leptin levels, with a trend towards increased adiponectin concentrations following personalized MD [137]. A study of 115 obese women (35–55 years) showed that, after two years of intervention with an MD and exercise program, adiponectin, hsCRP, IL-6, and TNF-α levels decreased in the group that lost less than 5% of initial weight, while resistin increased. In the groups that lost between 5% and 10% and more than 10% of initial weight, hsCRP, IL-6, and TNF-α levels decreased significantly, while resistin increased only in the group that lost more than 10% [138]. During pregnancy, a study of 99 women found that serum adiponectin levels decreased significantly from the first to the third trimester, confirming a low-grade inflammatory condition. However, women with greater adherence to the MD showed a smaller percentage reduction in adiponectin levels [139]. A cross-sectional analysis of baseline data from a study of 65 overweight and obese subjects found that a higher MD adherence score was associated with greater insulin sensitivity and higher levels of adiponectin and adipsin [140]. In a 12-month intervention involving 166 metabolically healthy obese subjects, adherence to the MD increased significantly, leading to a significant increase in adiponectin concentrations and a decrease in resistin concentrations [141].

In conclusion, the MD is shown to be effective in improving adiponectin levels, contributing to the management of metabolic health and the reduction of risks associated with metabolic syndrome and cardiovascular disease. 

**Table 2 nutrients-16-02436-t002:** Impact of the Mediterranean diet on adiponectin levels.

Author	Year	*n*	Study Design	Population	Duration	Adiponectin	Other Outcomes	Ref.
Pischon	2005	532	Cross-sectional	General population	N/A	↑	↓ glycemic index foods, omega-3, olive oil	[134]
Bédard	2014	70	Controlled intervention study	General population	4 weeks	↑		[130]
Spadafranca	2018	99	Cohort study	Pregnant women	During pregnancy	↓	↑ LDL cholesterol, or ↑ total cholesterol/HDL ratio	[139]
Sureda	2018	608	Cross-sectional	General population	N/A	↑	↓ leptin, ↓ TNF-α, ↓ PAI-1, ↑ hs-CRP	[129]
Gomez-Huelgas	2019	115	RCT	Obese women (35–55 years)	12 months	↓	↓ hsCRP, IL-6, ↓ TNF	[138]
Luisi	2019	18	RCT	Overweight/obese subjects	3 months	↑	↓ proinflammatory cytokines, ↑ IL-10	[136]
de Luis	2020	135	Interventional controlled trial	Obese patients with ADIPOQ gene polymorphism	12 weeks	↑	↑ lipid profiles, ↑ CRP	[133]
Bozzetto	2021	45	RCT	Type 1 diabetic patients	8 weeks	↑	↑ MUFA,↑ PUFA, ↓ SFA	[131]
Cobos-Palacios	2022	166	Interventional study	Metabolically healthy obese subjects	12 months	↑	↓ resistin, ↓ waist circumference, ↓ physical activity	[141]
Gioxari	2022	11	Pilot uncontrolled intervention trial	Overweight/obese men and women	N/A	↑	↑ body fat, ↑ waist circumference, ↑ leptin levels,	[137]
Sood	2022	65	Cross-sectional	Overweight and obese subjects	N/A	↑	↑ insulin, ↓ NF-κB, ↑ adipsin	[140]
de Luis	2023	133	Single-group, pre–post intervention design	General population	3–9 months	↑	↑ serum lipid profile	[135]
Scoditti	2023	N/A	Preclinical study	Human Simpson–Golabi–Behmel syndrome (SGBS) adipocytes and murine 3T3-L1 adipocytes	N/A	↑	↑ PPARγ expression and activity, ↓ JNK-mediated suppression, ↓ leptin-to-adiponectin ratio	[132]

Table 2 details the influence of the Mediterranean diet (MD) on adiponectin levels. The MD promotes high consumption of fruits, vegetables, fiber, unsaturated fats, whole grains, unrefined carbohydrates, a balanced consumption of dairy products and fish, and a reduced intake of saturated fatty acids and simple sugars. Numerous studies have shown that high adherence to this nutritional pattern is associated with increased levels of adiponectin. Abbreviations: CRP—C-reactive protein; HDL—high-density lipoprotein; hs-CRP—high-sensitivity C-reactive protein; IL-6—interleukin 6; IL-10—interleukin 10; JNK—c-Jun N-terminal kinases; LDL—low-density lipoprotein; MUFA—monounsaturated fatty acids; NF-κB—nuclear Factor kappa-light-chain-enhancer of activated B cells; PAI-1—plasminogen activator inhibitor-1; PPARγ—peroxisome proliferator-activated receptor gamma; PUFA—polyunsaturated fatty acids; RCT—randomized controlled trial; SFA—saturated fatty acids; TNF-α—tumor necrosis factor-alpha. ↑ indicates an increase; ↓ indicates a decrease.

### 4.3. The DASH Diet

The DASH (dietary approaches to stop hypertension) diet is a dietary pattern based on the consumption of vegetables, fruits, nuts, pulses, whole grain products, and low-fat dairy products, characterized by a low consumption of processed red meat, sugary drinks and high-sodium products. Similar to the MD, the DASH dietary approach is characterized by the same dietary indications, with the addition of limiting sodium intake. Trials analyzing this diet in relation to the serum concentration of adiponectin have obtained results superimposed on those obtained from the MD. Therefore, it is likely that the reason DASH is associated with an increased adiponectin concentration is due to the presence of bioactive components with strong anti-inflammatory properties, such as omega-3 fatty acids and polyphenols. 

The low sodium intake and anti-inflammatory components of the DASH diet, such as high consumption of fruit, vegetables, and low-fat dairy products, have shown promising effects on increasing serum adiponectin levels. Although individual studies, such as that of Saneei et al. [142], showed conflicting results regarding adiponectin, the overall evidence, including the meta-analysis by Soltani et al. [143], supports the beneficial role of the DASH diet in increasing adiponectin concentrations. These results suggest that the DASH diet could be a valid dietary approach to improve adiponectin levels.

### 4.4. Vegetarian Diet

A plant-based diet can positively influence serum adiponectin concentrations. Although intuitively a high consumption of vegetables may seem beneficial for adiponectin levels, this relationship is not yet fully demonstrated. From a functional point of view, a reduced intake of total and animal protein, which is characteristic of a vegetarian diet, could improve adipocyte function and, consequently, increase adiponectin levels due to the content of bioactive molecules typical of a plant-based diet [144].

Several recent studies (Table 3) have explored the effects of various diets on inflammatory biomarkers and adipokine levels, highlighting differences according to dietary habits and gender. A study of healthy, non-obese adults found significantly higher concentrations of adiponectin in women following a vegetarian diet compared to those following a conventional diet. However, this relationship was not observed in men [145]. In a cross-sectional analysis of 1986 middle-aged and elderly adults, a healthy plant-based diet (PBD) was associated with a more favorable inflammatory biomarker profile, with lower concentrations of C-reactive protein (CRP), interleukin-6 (IL-6), white blood cells (WBCs), neutrophils, and monocytes, and a lower leptin/adiponectin ratio [146]. Another cross-sectional study of 36 vegans and 36 omnivores found that the duration of the vegan diet appeared to influence the inflammatory profile, with trends towards differences in inflammatory biomarkers according to plasma levels of SFA or PUFA [147]. A cross-sectional study on 90 participants found no significant differences in leptin and adiponectin levels between vegetarians and non-vegetarians [148]. A study of 105 prepubertal children found that a lacto-ovo-vegetarian diet did not affect serum levels of adiponectin, visfatin, or omentin, although adiponectin/leptin and omentin/leptin ratios were significantly higher in vegetarians than in omnivores [149]. A randomized study of 53 healthy participants showed that in women the adiponectin level was significantly higher at the end of the trial on a vegan diet compared to a meat-rich diet, while in men, the adiponectin concentration remained stable [150]. 

Other recent studies have examined the impact of vegetarian diets on various health conditions, revealing distinct effects on inflammatory markers, adipokines, and general health profiles. A study of women with polycystic ovary syndrome (PCOS) showed that those consuming an Indian vegetarian diet had higher levels of proinflammatory markers and lower levels of anti-inflammatory markers than their non-vegetarian counterparts [151]. In a 24-week randomized trial of patients with type 2 diabetes, a low-calorie vegetarian diet significantly increased adiponectin and reduced leptin compared to a conventional diabetic diet. This diet showed a greater ability to improve insulin sensitivity, reduce visceral fat and improve plasma concentrations of adipokines and markers of oxidative stress, especially when combined with exercise [152]. In another study, a vegetarian diet compared to a Mediterranean diet for three months showed no significant difference in adiponectin levels but resulted in a significant reduction in the leptin/adiponectin ratio and anthropometric parameters [153]. In a population of South Asian origin in the United States, higher scores on the healthy plant diet index were associated with lower levels of glycated hemoglobin, higher adiponectin, and smaller areas of visceral and pericardial fat [154].

In conclusion, the studies reviewed highlight the potential benefits of vegetarian and plant-based diets on various health parameters, particularly inflammatory biomarkers and adipokine levels. These diets have shown promise in increasing adiponectin levels, improving insulin sensitivity and reducing markers of inflammation and oxidative stress. However, the effects may vary depending on gender, specific health conditions, and duration of adherence to the diet. Although vegetarian diets appear to be useful in improving some health markers, further research is needed to fully understand their impact and optimize dietary recommendations for different populations.

**Table 3 nutrients-16-02436-t003:** Impact of vegetarian diet on adiponectin levels.

Author	Year	*n*	Study Design	Population	Duration	Adiponectin	Other Outcomes	Ref.
Kahleova	2011	72	RCT	Patients with type 2 diabetes	24 weeks	↑	↑ insulin sensitivity, ↑ oxidative stress markers, ↓ leptin, ↓ visceral fat	[152]
Ganie	2019	464	Observational case–control study	Women with PCOS	N/A	↓	↑ proinflammatory markers, ↓ anti-inflammatory markers	[151]
Baden	2019	831	Cross-sectional	South Asians in USA	N/A	↑	↓ glycated hemoglobin	[154]
Dinu	2020	118	RCT, crossover	General population (Clinically healthy subjects)	3 months	↑	↓ leptin, ↑ anthropometric parameters	[153]
Menzel	2020	72	Cross-sectional	Vegans vs. omnivores	N/A	=	Variation in inflammatory biomarkers	[147]
Vučić Lovrenčić	2020	76	Case–control	Healthy, non-obese adults	N/A	↑ (females)	↑ beta-cell function in vegetarians, = in men	[145]
Ambroszkiewicz	2021	105	Cross-sectional	Prepubertal children	N/A	↑	leptin ratio, ↑ omentin- leptin ratio	[149]
Lederer	2022	53	RCT	Healthy women	4 weeks	↑		[150]
Kharaty	2023	1986	Cross-sectional	Middle age vs. elderly adults	N/A	↑	↓ CRP, ↓ IL-6, ↓ WBC, ↓ neutrophils, ↓ monocytes, ↓ leptin	[146]
García-Maldonado	2023	90	Cross-sectional	Vegetarians vs. non-vegetarians	N/A	=		[148]

Table 3 details the influence of a vegetarian diet on adiponectin levels. This diet is characterized by high consumption of vegetables, fruits, and plant-based foods, with a reduced intake of total and animal protein. Various studies have explored the effects of vegetarian diets on inflammatory biomarkers and adipokine levels, highlighting differences based on dietary habits and gender. Abbreviations: CRP—C-reactive protein; IL-6—interleukin 6; PCOS—polycystic ovary syndrome; RCT—randomized controlled trial; WBC—white blood cell. ↑ indicates an increase; ↓ indicates a decrease; = denotes that the levels of adiponectin have not changed significantly.

### 4.5. Ketogenic Diet

The ketogenic diet is characterized by very low carbohydrate intake, moderate protein consumption, and high fat intake, which induce the production of ketone bodies (acetoacetate, β-hydroxybutyrate, and acetone). Initially used in the 1920s as a treatment for drug-resistant epilepsy, this dietary approach has gained popularity in the last two decades. Recent evidence suggests that ketone bodies not only serve as energy substrates, but also play multiple roles in enhancing mitochondrial function, acting as signaling molecules, and strengthening endogenous antioxidant defenses [155].

Most studies (Table 4) have shown various positives of the ketogenic diet on adiponectin levels. For example, in healthy young individuals, the transition from a habitual high-carbohydrate diet to a very low-carbohydrate, high-fat diet (VLCHF) while maintaining regular physical activity for 12 weeks led to beneficial changes in serum concentrations of adiponectin and leptin, as well as favorable changes in body weight and fat mass [156]. Long-term adherence to the ketogenic diet has been shown to increase serum levels of adiponectin in mice [157]. In children with drug-resistant epilepsy treated with the ketogenic diet, serum levels of adiponectin and omentin-1 were significantly higher and vaspin levels lower than those treated with valproic acid alone and healthy controls [158]. An eight-week intervention with an ultra-low-calorie ketogenic diet (VLCKD) resulted in significant weight loss in participants, decreased production of proinflammatory cytokines, increased serum adiponectin levels, and improved metabolic profile [159]. A significant reduction in insulin and leptin levels, together with a significant increase in adiponectin and neuropeptide Y (NPY) [160], was observed over the course of 12 weeks on the ketogenic diet. In participants without complications from type 2 diabetes (T2D) or cardiovascular disease (CVD), a 12-month ketogenic diet led to a significant increase in adiponectin levels, independent of weight loss [161]. Furthermore, in a study that compared the effects of a ketogenic diet and a low-calorie diet on the metabolic parameters of obese children and adolescents, adiponectin levels increased significantly only in the ketogenic group after six months [162]. However, in children with glucose transporter 1 deficiency syndrome, a 12-week ketogenic diet did not change the levels of biomarkers of inflammatory activity and adipose tissue, including high-sensitivity C-reactive protein (hs-CRP), TNF-α, IL-6, leptin, adiponectin, and free fatty acids [163].

In conclusion, although the ketogenic diet has demonstrated several beneficial effects on adiponectin levels and metabolic health, these effects are probably attributed to the significant calorie restriction inherent in the diet. Furthermore, there are considerable doubts about the long-term sustainability of the ketogenic diet, given its highly restrictive nature and potential nutritional deficiencies. 

### 4.6. Specific Factors Affecting Adiponectin Levels

Several studies both preclinical (Table 5) and in humans (Table 6) have evaluated the effects of food on adiponectin levels. Studies have mainly focused on the influence of specific elements, such as omega-3 fatty acids, probiotics, and alcohol, on adiponectin levels.

#### 4.6.1. Effects of Omega-3 Fatty Acids, Probiotics, and Alcohol on Adiponectin Levels in Preclinical Studies

In preclinical studies, omega-3 supplementation showed positive effects on adiponectin levels. Perilla oil, an omega-3-rich vegetable oil, and omega-3 oil monotherapy led to increased adiponectin levels and partial improvements in metabolic profiles [164,165]. In studies on 3T3-L1 adipose cells, DHA supplementation increased adiponectin secretion compared to untreated hypoxic cells, and also reduced proinflammatory markers such as IL-6 and MCP-1, as well as leptin [166]. Besides PUFA, probiotics have also been shown to positively influence adiponectin levels in preclinical studies. Administration of *Lactobacillus rhamnosus GG* (*LGG*) for 13 weeks to mice fed a high-fat diet showed a significant increase in adiponectin production in adipose tissue [167]. An experiment on 55 male C57BL/6J mice fed a glucose- and fat-rich diet showed that supplementation for 8 weeks with *Lactobacillus paracasei JY062* increased adiponectin levels [168]. In rats fed a high-fat diet, treatment with yogurt fermented by *Lactobacillus acidophilus NX2-6 (LA-Y)* resulted in higher serum levels of adiponectin and glycerol than the high-fat diet group [169]. In contrast, alcohol exposure showed no significant effects on adiponectin levels. In a study of rats exposed to ethanol, no significant differences were observed in adiponectin levels compared to control rats, nor in albumin, blood glucose, triglycerides (TG), total cholesterol (TC), or oxidized LDL-cholesterol (oxLDL-C) levels [170].

**Table 5 nutrients-16-02436-t005:** Impact of omega-3 fatty acids, probiotics, and alcohol on adiponectin levels in preclinical studies.

Author	Year	*n*	Population	Duration	Treatment	Adiponectin	Other Outcomes	Ref.
Kim	2012	7–8 mice per group	Rats	13 weeks	*Lactobacillus rhamnosus GG (LGG)*	↑	↑ insulin sensitivity, ↓ adiposity, ↑ AMPK activation, ↑ fatty acid oxidation in liver and muscle, ↓ gluconeogenesis in liver, ↑ GLUT4 expression in muscle	[167]
Thomas	2020	40	Rats	16 weeks	ω-3-FA	↑	↓ obesity markers, ↓ inflammation, ↑ gut health	[164]
Refaat	2022	40	Rats with metabolic dysfunction-associated fatty liver disease (MAFLD)	4 weeks	ω-3-FA and/or vitamin D3	↑ (Partial)	↑ metabolic profile	[165]
Younes	2022	N/A	3T3-L1 adipocytes	N/A	DHA	↑	↓ proinflammatory markers (IL-6, MCP-1) and leptin	[166]
Tang	2023	N/A	Mice fed a high-fat diet	12 weeks	*L. acidophilus NX2-6*	↑	Improved lipid profile, ↓ fasting blood glucose levels, ↑ insulin sensitivity, ↑ fatty acid oxidation, ↓ de novo lipogenesis, ↑ insulin signaling, ↓ gluconeogenesis, ↓ proinflammatory cytokines, ↑ glycolysis and gluconeogenesis, ↑ fat browning, ↑ mitochondrial biogenesis, ↑ energy expenditure	[169]
Kwon	2024	24	Adult Wistar rats	N/A	Moderate ethanol exposure	=	↓ CRP levels, trends towards decreased HMGB-1, TNF-a, and 8-oxo-dG levels; upregulated Sod gene expression in colon	[170]
Su	2024	55	Male C57BL/6J mice	11 weeks	*Lactobacillus paracasei JY062*	↑	↓ TC, TG, LDL-C, leptin, insulin, and FFA; ↑ HDL-C and GLP-1; improved gut microbiota balance	[168]

Table 5 details the preclinical studies that evaluated the influence of specific nutrients, such as omega-3 fatty acids, probiotics, and alcohol, on adiponectin levels. Abbreviations: ACOX—acyl-CoA oxidase; ACC—acetyl-CoA carboxylase; AMPK—AMP-activated protein kinase; CPT1—carnitine palmitoyltransferase 1; FAS—fatty acid synthase; G6Pase—glucose-6-phosphatase; GLUT4—glucose transporter type 4; HFL—high-fat diet-fed LGG-treated mice; HFP—high-fat diet-fed PBS-treated mice; LGG—*Lactobacillus rhamnosus* GG; NDL—normal diet-fed LGG-treated mice; NDP—normal diet-fed PBS-treated mice; PEPCK—phosphoenolpyruvate carboxykinase; PPAR-α—peroxisome proliferator-activated receptor alpha; SREBP1c—Sterol regulatory element-binding protein 1c; ω-3-FA—omega-3 fatty acid. ↑ indicates an increase; ↓ indicates a decrease; = denotes that the levels of adiponectin have not changed significantly.

#### 4.6.2. Effects of Omega-3 Fatty Acids, Probiotics, and Alcohol on Adiponectin Levels in Human Studies

In human studies, omega-3 supplementation showed generally positive results on adiponectin levels. In patients with NAFLD, supplementation with Camelina sativa oil, an omega-3-rich vegetable oil, caused a significant increase in serum adiponectin concentration [171]. In peri/postmenopausal women with an increased risk of breast cancer, weight loss greater than 10% associated with w-3-FA showed significant improvements in biomarkers, including adiponectin [172]. Furthermore, although a small study found no significant differences in changes in adiponectin levels between the EPA+DHA and control groups [173], a non-significant increase in adiponectin levels was observed after 28 days of fish oil supplementation [174]. Another cross-sectional study examined the effect of dietary ω-6:ω-3 FA ratio on individuals with a BMI ≥ 18.5 kg/m^2^, finding that a high ω-6:ω-3 ratio is positively associated with reduced adiponectin, excessive adiposity, increased waist circumference, and worsened metabolic profile, including insulin and HOMA-IR [175].

Probiotic administration also showed effects on adiponectin levels. In a study of 86 patients with diabetic nephropathy undergoing hemodialysis treatment, the serum concentration of adiponectin was reduced in the observation group, treated with capsules containing *Lactobacillus acidophilus*, *Lactobacillus casei*, and *Bifidobacterium*, compared to the control group [176]. In contrast, the probiotic *Lactobacillus gasseri SBT2055 (LG2055)* significantly increased adiponectin levels, reducing abdominal adiposity, body weight, and other body measures in adults with obesity tendencies [177].

The effect of alcohol on adiponectin levels shows a non-linear association, with moderate drinkers being associated with higher adiponectin concentrations than non-drinkers. In a study of 532 men, moderate alcohol drinkers had significantly higher adiponectin levels, while high glycemic load was associated with lower adiponectin levels [134].

**Table 6 nutrients-16-02436-t006:** Impact of omega-3 fatty acids, probiotics, and alcohol on adiponectin levels in human studies.

Author	Year	*n*	Study Design	Population	Duration	Treatment	Adiponectin	Other Outcomes	Ref.
Pischon	2005	532	Cross-sectional	Male participants of HPFS	N/A	Dietary factors	↑ Alcohol intake		[134]
Kadooka	2010	87	RCT	Adults	12 weeks	*Lactobacillus gasseri SBT2055*	↑	↓ abdominal fat, ↓ BMI, ↓ waist, ↓ hip circumference	[177]
Torres-Castillo	2018	170	Cross-sectional	Individuals with a BMI ≥ 18.5 kg/m^2^	N/A	Dietary ω-6:ω-3 FA ratio	↓	↑ excessive adiposity, ↑ waist circumference, ↓ metabolic profile (insulin and HOMA-IR)	[175]
Fabian	2021	46	RCT	Peri/postmenopausal women at increased risk for breast cancer	6–12 months	ω-3-FA (+ weight loss)	↑	↓ body weight, ↓ leptin, ↓ proinflammatory markers	[172]
Pauls	2021	30	RCT	Women with obesity	4 weeks	ω-3-FA	↑	= inflammatory markers, altered monocyte bioenergetics	[174]
Rausch	2021	32	RCT	Older adults with obesity	8 weeks	EPA+DHA	=	↓ reduction in leptin-to-adiponectin ratio; associations with IL-1β and TNF-α	[173]
Musazadeh	2022	46	RCT	NAFLD patients	12 weeks	PUFA	↑	↓ BMI, ↓ waist circumference, ↓ triglycerides, ↓ TC, ↓ LDL-c,↓ ALT	[171]
Zhang	2023	86	RCT	Diabetic hemodialysis patients	12 weeks	*L. acidophilus, L. casei, Bifidobacterium*	↓	↑ serum ghrelin, ↑ nutrient intake, ↓ serum creatinine, ↓ fasting blood glucose, ↓ HOMA-IR, ↓ inflammatory markers	[176]

Table 6 shows the preclinical studies that evaluated the influence of specific nutrients, such as omega-3 fatty acids, probiotics, and alcohol, on adiponectin levels. Abbreviations: ACOX—acyl-CoA oxidase; ACC—acetyl-CoA carboxylase; ACOX—acyl-CoA oxidase; ALT—alanine aminotransferase; AMPK—AMP-activated protein kinase; BMI—body mass index; CPT1—carnitine palmitoyltransferase 1; DHA—docosahexaenoic acid; EPA—eicosapentaenoic acid; FAS—fatty acid synthase; G6Pase—glucose-6-phosphatase; GLUT4—glucose transporter type 4; HFD—high-fat diet; HOMA-IR—homeostatic model assessment for insulin resistance; LDL-c—low-density lipoprotein cholesterol; LGG—Lactobacillus rhamnosus GG; NAFLD—nonalcoholic fatty liver disease; ND—normal diet; PEPCK—phosphoenolpyruvate carboxykinase; PPAR-α—peroxisome proliferator-activated receptor alpha; PUFA—polyunsaturated fatty acid; RCT—randomized controlled trial; SREBP1c—sterol regulatory element-binding protein 1c; TC—total cholesterol; ω-3-FA—omega-3 fatty acid; ω-6-FA—omega-6 fatty acid. ↑ indicates an increase; ↓ indicates a decrease; = denotes that the levels of adiponectin have not changed significantly.

#### 4.6.3. Summary of Reviews on Adiponectin and Various Factors

Other review studies have evaluated the effect of various nutrients on adiponectin levels, showing interesting and sometimes conflicting results. In obese animal models, omega-3-containing lipids increased serum adiponectin levels [178]. Overall, while many studies indicate a potential benefit of omega-3 in increasing adiponectin levels, there is also evidence of variability in effects depending on the population studied and the doses used. A review of 52 studies showed that consumption of more than 2000 mg per day of omega-3 for more than 10 weeks was associated with a significant increase in plasma adiponectin levels [179]. A 14–60% increase in adiponectin levels was observed with daily intake of fish or omega-3 supplements [180]. Another review of 60 RCTs on 3845 patients with T2DM confirmed the increase in adiponectin levels with fish oil intake [181]. Other studies examined the effect of probiotics on adiponectin levels. A review of 60 RCTs on 3845 patients with type 2 diabetes indicated that probiotics increase adiponectin levels, albeit to a lesser extent than fish oil [181]. A review of 26 RCTs (*n* = 1536) showed that supplementation with probiotics/symbiotics is associated with a tendency to increase adiponectin levels, particularly in patients with type 2 diabetes, metabolic syndrome, and prediabetes [182]. However, another review of 32 RCTs on subjects with prediabetes and type 2 diabetes found no significant effects of probiotic and symbiotic supplementation on serum adiponectin levels [183]. These studies suggest that although there is evidence for a positive effect of omega-3 and probiotics on increasing adiponectin levels, results vary widely depending on the population studied, dosages used, and duration of interventions. Therefore, further research is needed to better understand the impact of these nutrients on adiponectin levels and to identify optimal conditions for their use.

### 4.7. How Diet and Exercise Affect Adiponectin Levels

Diet and exercise synergistically modulate adiponectin levels, with higher blood concentrations observed in individuals who engage in more physical activity, potentially due to improved insulin sensitivity [165]. Several studies highlight the benefits of combining different diets with regular physical activity. For example, a MD with calorie restriction and aerobic exercise led to a 15% weight loss and a 48% increase in adiponectin [184]. The production of adiponectin by muscle cells suggests regulatory functions beyond insulin-sensitive tissue, influenced by factors such as diet, muscle type, and exercise intensity [185]. Dietary fiber intake has been shown to increase adiponectin levels due to its beneficial effects on the gut microbiota, which can enhance anti-inflammatory properties and improve insulin sensitivity. Several studies have shown that dietary fiber positively influences the production of short-chain fatty acids, which can stimulate adiponectin secretion. For example, a study by Zeinabi et al. [186] found that soluble fiber had beneficial effects on adiponectin levels, highlighting its role in modulating cardiometabolic risk factors. Another study by Bozzetto et al. [131] demonstrated that a diet rich in low glycemic index carbohydrates and fiber significantly increased postprandial adiponectin levels in patients with type 2 diabetes compared to a diet rich in monounsaturated fatty acids.

Exercise has been shown to reduce inflammatory markers and increase adiponectin in overweight and obese postmenopausal women [187]. In a combined 13-week lifestyle intervention, adiponectin levels decreased along with other markers [188]. Comparison of moderate- and high-intensity interval training (HIIT) revealed different effects on metabolic health, with HIIT increasing adiponectin [189]. Acute and chronic exercise interventions show a different impact on adiponectin, with long-term HIIT significantly increasing its levels [190,191,192,193]. In postmenopausal women with obesity, combined diet and exercise interventions significantly increased adiponectin [194]. Strength training acutely increased adiponectin in young males [195], whereas 12-week exercise programs did not significantly alter adiponectin in men with metabolic syndrome [196]. Aerobic exercise and HIIT effectively increased adiponectin in overweight and obese women [197]. Furthermore, HIIT can increase adiponectin in breast milk after exercise [198]. Aerobic interval and resistance training significantly improved adiponectin in sedentary men with metabolic syndrome [199], and aerobic exercise shows promise for increasing adiponectin in overweight and obese children and adolescents [200]. Calorie restriction combined with exercise increased adiponectin in postmenopausal women, in contrast to calorie restriction alone [201]. No relationship was found between physical activity and adipokine concentrations in women with PCOS [202], although moderate aerobic exercise improved adiponectin in this group [203]. However, other reviews suggest that aerobic exercise alone may not significantly influence adiponectin levels [204]. Recent studies have indicated that different forms of adiponectin have distinct roles in metabolic health. Lara-Castro et al. [205] showed that it is HMW adiponectin that correlates primarily with increased insulin sensitivity, reduced abdominal fat, and elevated basal lipid oxidation. The study suggested that the ratio of adiponectin HMW to total adiponectin is a significant indicator of metabolic health, underlining the importance of HMW adiponectin in explaining the cluster of metabolic syndrome traits. Similarly, Lara-Castro et al. [206] observed that the decrease in total adiponectin in African Americans was mainly due to a reduction in LMW and trimer forms, whereas HMW levels did not differ significantly between races. They found that HMW was highly correlated with multiple metabolic syndrome traits in Europeans, while LMW and trimer forms were more strongly associated with these traits in African Americans.

In conclusion, regular physical activity contributes significantly to increasing adiponectin levels, which are crucial for metabolic health. Types of exercise such as aerobic activity, HIIT and strength training have shown positive effects on adiponectin concentrations. The consistent finding of studies is that physical activity improves insulin sensitivity and increases adiponectin levels, highlighting the importance of incorporating regular exercise into the daily routine to improve metabolic health. However, for optimal results, these benefits are often more pronounced when combined with dietary changes.

## 5. Discussion

Adiponectin is one of the most studied adipokines to date. First described in the mid-1990s, analysis of its regulation and physiological effects on its biogenesis has proved extremely important and has improved our understanding of the mechanisms involved in systemic metabolic homeostasis [207]. The main functions of adiponectin are classified as insulin-sensitizing, anti-fibrotic (found in many tissues, particularly kidney, liver, and adipose tissue), anti-apoptotic, and anti-inflammatory, and high levels of the hormone protect against liver and kidney fibrosis [4]. On a systemic level, adiponectin is able to regenerate tissue: studies on mice have shown that in kidney-deficient and adiponectin-deficient specimens, it causes irreversible failure; on the contrary, a rapid recovery of kidney function is observed with an overexpression of adiponectin [77]. Another example of the regenerative effects of adiponectin is observed in pancreatic β-cells, where adiponectin promotes β-cell reconstitution after apoptotic damage [208,209]. 

Given the significant role that adiponectin plays in metabolic homeostasis, inflammation reduction, and tissue regeneration, it is critical to understand which nutritional and non-nutritional therapies can increase adipokine levels, particularly adiponectin. Dietary interventions, physical activity and pharmacological approaches that effectively increase adiponectin levels could potentially offer therapeutic benefits for conditions such as insulin resistance, type 2 diabetes, and chronic inflammation.

In this review, we highlighted the impact of various dietary patterns, specific nutrients, and physical activities on adiponectin levels. Calorie restriction alone has been shown to positively influence adiponectin concentrations. Studies have shown that body weight reduction through low-calorie diets and exercise can significantly increase adiponectin levels, improving insulin sensitivity and reducing inflammation [125,126]. However, although caloric restriction is beneficial, it may not be sufficient on its own to optimize adiponectin levels and improve metabolic health comprehensively. Dietary patterns such as the MD and the DASH diet have been associated with increased adiponectin levels. The MD, characterized by a high consumption of fruit, vegetables, whole grains, and healthy fats, has been shown to improve insulin sensitivity and reduce inflammation, mainly due to its rich content of omega-3 fatty acids and polyphenols [130,134]. Similarly, the DASH diet, which emphasizes low-fat fruits, vegetables, and dairy products while limiting sodium intake, has also shown beneficial effects on serum adiponectin levels, contributing to improved metabolic health [143]. Specific nutrients such as omega-3 fatty acids, found in fish oil, have been linked to a significant increase in adiponectin levels due to their anti-inflammatory properties and improved lipid metabolism [179,181]. Furthermore, probiotics such as *Lactobacillus rhamnosus GG* have been shown to increase adiponectin levels, especially when combined with a fiber-rich diet [167]. Regular exercise, including aerobic and resistance training, has been consistently associated with higher adiponectin levels, increased insulin sensitivity, and reduced systemic inflammation in various populations, including those with metabolic syndrome and type 2 diabetes [126,127]. Overall, these interventions demonstrate potential strategies to increase adiponectin levels to promote metabolic health and prevent related diseases. 

Whether adiponectin-targeted interventions directly improve metabolic health or whether changes in adiponectin levels are simply indicative of broader lifestyle changes remains an important consideration. Although increasing adiponectin levels through diet and exercise appears promising, it is critical to determine whether these changes are causal in improving metabolic health or simply related to overall healthier lifestyle practices. Understanding the direct role of adiponectin modulation in metabolic improvement can help refine therapeutic strategies and ensure that interventions are targeted to the most effective pathways for disease prevention and management [210,211]. The correlation between dietary components and adiponectin expression is still unclear and requires further research. Currently, insufficient data are available to confirm a direct relationship between individual dietary elements and the expression of this adipokine. Several questions concerning adiponectin remain unanswered, the post-transcriptional mechanism regulating adiponectin secretion being in the spotlight given the enormous abundance of mRNA. The metabolic state of adipocytes, in particular the functional integrity of their mitochondria, has a significant impact on adiponectin production. However, the exact mechanism of this process remains unclear. Furthermore, adiponectin plays a role in the physiological responses of cardiac, renal, and hepatic cells. Understanding how these interactions occur could reveal new insights into adipose tissue physiology and reveal the importance of the systemic signaling axis in maintaining metabolic homeostasis during obesity and insulin resistance. 

Although this review has collected and analyzed a wide range of studies on adiponectin and its role in metabolic health, several limitations must be emphasized. First, the variability in study designs, populations, and methods of measuring adiponectin levels makes it difficult to draw definitive conclusions in all studies. Furthermore, many studies are based on animal models, which may not fully replicate human physiology and responses. The long-term effects of interventions aimed at increasing adiponectin levels are also not well documented, requiring further longitudinal studies to establish a lasting impact on metabolic health. Finally, the interplay between genetic factors and lifestyle interventions in regulating adiponectin levels remains poorly explored, highlighting the need for more comprehensive and multifaceted research approaches.

## 6. Conclusions

Adiponectin plays a critical role in maintaining systemic metabolic homeostasis through its insulin-sensitizing, antifibrotic, anti-apoptotic, and anti-inflammatory properties. This review highlighted the potential of various dietary patterns, specific nutrients, and physical activities to increase adiponectin levels and improve metabolic health. Although calorie restriction, the Mediterranean diet, the DASH diet, omega-3 fatty acids, probiotics, and regular exercise have shown promise in improving adiponectin concentrations, further research is needed to determine the causal relationships and long-term effects of these interventions. The effectiveness of adiponectin-targeted interventions in directly improving metabolic health remains a crucial question. Understanding the direct role of adiponectin modulation may refine therapeutic strategies and ensure effective pathways for disease prevention and management. Comprehensive and multifaceted research approaches are needed to explore the complex interactions between genetics, lifestyle interventions, and adiponectin regulation. By addressing current knowledge gaps, future studies may provide insights into the physiological roles of adiponectin and its potential as a therapeutic target for metabolic diseases.

## Figures and Tables

**Figure 1 nutrients-16-02436-f001:**
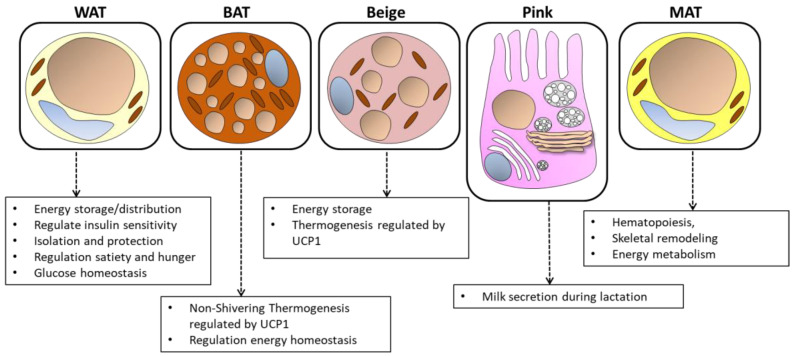
The structure and functions of white, brown, beige, pink, and marrow adipose tissue. Energy storage, endocrine communication, and insulin sensitivity are the main activities that characterize white adipose tissue (WAT). Thermogenesis and energy expenditure instead characterize brown adipose tissue (BAT). Beige adipocytes, present in WAT, also contribute to thermogenesis. Pink adipocytes are formed during pregnancy and lactation and play a role in milk secretion. Medullary adipose tissue (MAT) is located in the bone marrow and participates in local and systemic metabolic processes. This figure illustrates the cellular characteristics and distinct physiological roles of each type of adipose tissue.

**Figure 2 nutrients-16-02436-f002:**
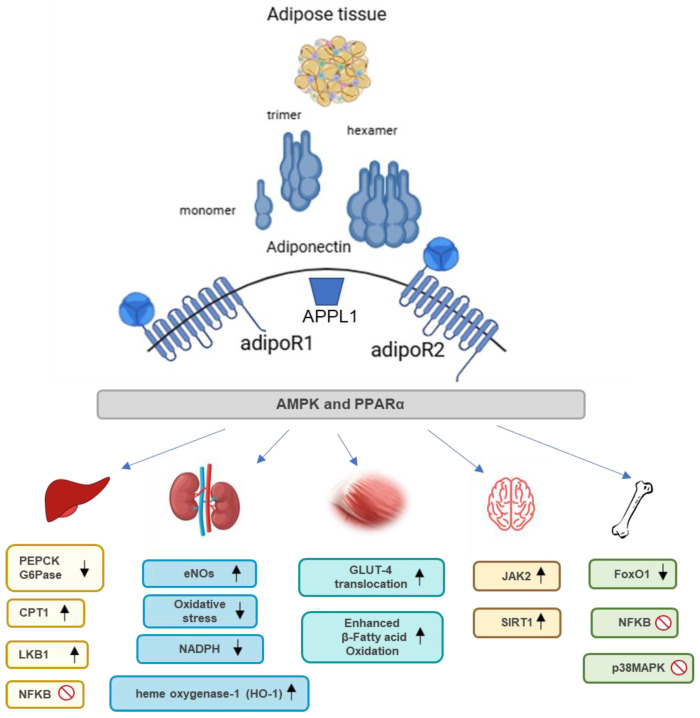
Structure, receptors, and function of adiponectin in various organs. Adiponectin, produced by adipose tissue, exists in various forms, such as monomers, trimers, and hexamers. It acts through its receptors, adipoR1 and adipoR2, to influence multiple metabolic processes. Activation of adipoR1 mainly promotes mitochondrial biogenesis and glucose uptake through the AMPK and p38MAPK pathways. AdipoR2 activation is involved in fatty acid oxidation through the PPARα pathway and provides cytoprotective and anti-inflammatory effects through modulation of NF-κB. Collectively, these pathways contribute to the regulation of energy metabolism and inflammatory responses in various tissues, including muscle, liver, kidney, brain, and bone. ↑ indicates an increase or upregulation; ↓ indicates a decrease or downregulation. Abbreviations: CPT1: Carnitine palmitoyltransferase I; eNOS: Endothelial nitric oxide synthase; FoxO1: Forkhead box protein O1; G6Pase: Glucose-6-phosphatase; GLUT-4: Glucose transporter type 4; JAK2: Janus kinase 2; LKB1: Liver kinase B1; NADPH: Nicotinamide adenine dinucleotide phosphate; NFκB: Nuclear factor kappa-light-chain-enhancer of activated B cells; PEPCK: Phosphoenolpyruvate carboxykinase; p38MAPK: p38 mitogen-activated protein kinase; SIRT1: Sirtuin 1.

**Table 1 nutrients-16-02436-t001:** Effect of adiponectin on human diseases.

Disease	Role of Adiponectin	Mechanisms of Action	References
Cardiovascular diseases	Inversely correlated with coronary heart disease risk; lower levels linked to hypertension and cardiomyopathy; elevated in heart failure	Enhances NO synthesis, attenuates endothelial adhesion molecules	[60,96,97]
Diabetes and insulin resistance	Negatively correlated with type 2 diabetes prevalence	Improves insulin sensitivity and glucose tolerance, reduces TG in adipose tissue, activates PPAR-α receptor, AMPK cascade	[68]
Metabolic syndrome	Low levels linked to increased risk of metabolic syndrome; inversely related to waist circumference, visceral fat, and triglycerides	Improves lipid metabolism, anti-inflammatory effects	[98]
Chronic liver disease and NAFLD	Elevated in chronic liver disease; reduced in NAFLD; associated with liver inflammation and damage	Activates AMPK, reduces hepatic glucose production, enhances fatty acid oxidation	[99,100]
Alzheimer’s disease	Lower levels may accelerate progression and cognitive decline; protective neuroprotective effects	Enhances insulin signaling, reduces amyloid-beta deposition and tau hyperphosphorylation	[101,102]
Cancer	Lower levels associated with higher risk of breast, colorectal, and other cancers; influences cell proliferation and apoptosis	Suppresses cell proliferation, promotes apoptosis via AMPK and other pathways	[103,104,105,106]

Table 1 summarizes the various roles of adiponectin in human diseases, including cardiovascular diseases, diabetes, insulin resistance, metabolic syndrome, chronic liver disease, Alzheimer’s disease, and cancer. Abbreviations: AMPK—AMP-activated protein kinase; HDL—high-density lipoprotein; hs-CRP—high-sensitivity C-reactive protein; IL-6—interleukin 6; LDL—low-density lipoprotein; PAI-1—plasminogen activator inhibitor-1; TNF-α—tumor necrosis factor-alpha.

**Table 4 nutrients-16-02436-t004:** Effects of the ketogenic diet on adiponectin levels.

Author	Year	*n*	Study Design	Population	Duration	Adiponectin	Other Outcomes	Ref.
Partsalaki	2012	58	RCT	Obese children and adolescents	6 months	↑	↓ weight, ↓ fat mass, ↓ waist circumference, ↑ insulin	[162]
Bertoli	2015	10	Longitudinal study	Children with GLUT1 deficiency syndrome	12 weeks	=	↓ insulin, ↓ HOMA-IR, ↑ QUICKI	[163]
Hu, T	2015	148	RCT	Participants without T2D or CVD complications	12 months	↑	↓ intercellular adhesion molecule-1 (ICAM-1)	[161]
Mohorko	2019	35	Uncontrolled intervention	General population—sedentary obese	12 weeks	↑	↓ insulin, ↓ leptin, ↑ NPY	[160]
Monda, V	2020	20	RCT	General population—Obese subjects	8 weeks	↑	↓ weight loss, ↓ pro-inflammatory cytokines	[159]
Cipryan	2021	24	Non-randomized, parallel design.	Healthy young individuals	12 weeks	↑	↑ leptin, ↑ body weight, ↑ fat mass	[156]
Widiatmaja	2021	N/A	Preclinical study	Rats	Long-term	↑		[157]
Chyra	2022	72	Cross-sectional	Children with drug-resistant epilepsy	>3 months for keto group	↑	↑ omentin-1, ↓ vaspin	[158]

Table 4 details the influence of the ketogenic diet on adiponectin levels. This diet is characterized by very low carbohydrate intake, moderate protein consumption, and high fat intake, which induce the production of ketone bodies. Studies have shown various benefits of the ketogenic diet on adiponectin levels and metabolic health. Abbreviations: CVD—cardiovascular disease; GLUT1—glucose transporter 1; HOMA-IR—homeostatic model assessment of insulin resistance; ICAM-1—intercellular adhesion molecule 1; IL-6—interleukin 6; NPY—neuropeptide Y; QUICKI—quantitative insulin sensitivity check index; RCT—randomized controlled trial; T2D—type 2 diabetes; TNF-α—tumor necrosis factor-alpha; hs-CRP—high-sensitivity C-reactive protein. ↑ indicates an increase; ↓ indicates a decrease; = denotes that the levels of adiponectin have not changed significantly.

## Data Availability

Data are available from the authors upon reasonable request. Data available on request due to restrictions for privacy reasons.

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
