# Peer review of "The Role of Adipose Tissue and Nutrition in the Regulation of Adiponectin"

_nutrients, 2024, doi:10.3390/nu16152436_

Round 1

Reviewer 1 Report

Comments and Suggestions for Authors

The paper entitled "The Role of Adipose Tissue and Nutrition in the Regulation of Adiponectin" is interesting and well organized. Nevertheless I have on major comment and several minor comments: Major: The paper needs to be revised and edited by a scientific grammar profesional editing service in order to improve its quality and readability. The paper is full of grammar mistakes and typos Minor: The authors should comment the effect of bariatric surgery on adiponectin levels (in section 3.2.2) Lines 224 to 228 it describes contradictory facts?? Please clarify Line 437 define BMD Line 540-41 please justify the fact that fiber increases adiponectin levels Line 822 high-intensity interval training = (HIIT). Please abreviate

Comments on the Quality of English Language

The paper needs to be revised and edited by a scientific grammar profesional editing service in order to improve its quality and readability

Author Response

Dear Reviewer,

First of all, we would like to thank you for the valuable impulses that allowed us to improve the quality of the manuscript. All changes made are highlighted by yellow color, in the revised version of the manuscript, to facilitate the review process. Hoping that we have satisfied your requests as much as possible, we kindly ask you to re-evaluate our paper. 

The Authors

Reviewer 1

Comments and Suggestions for Authors

The paper entitled "The Role of Adipose Tissue and Nutrition in the Regulation of Adiponectin" is interesting and well organized. Nevertheless I have on major comment and several minor comments: 

Dear Reviewer,

Thank you for your comments and suggestions. We appreciate your positive feedback on our paper. We have addressed your major comment and made the necessary revisions to improve the clarity and organization. Additionally, we have taken your minor comments into consideration and adjusted the manuscript accordingly.

Major: The paper needs to be revised and edited by a scientific grammar professional editing service in order to improve its quality and readability. The paper is full of grammar mistakes and typos 

Thank you for your important comment. We have reviewed the document and it was edited by a professional native English speaker to improve quality and readability. We hope that it now meets the required standards. Thank you again for your valuable feedback.

Minor: 

The authors should comment on the effect of bariatric surgery on adiponectin levels (in section 3.2.2) Lines 224 to 228 it describes contradictory facts?? Please clarify.

Thank you for your comment. We have revised lines 224-228 to better clarify the concept and resolve contradictions. In addition, we have added a comment on the effect of bariatric surgery on adiponectin levels in section 3.2.2.

Line 437 define BMD Line 540-41 please justify the fact that fiber increases adiponectin levels 

Thank you for your comments and suggestions. We appreciate your feedback and have made the following revisions to address your concerns:

  • We have added a definition for Bone Mineral Density (BMD) to clarify its meaning.
  • We have included an explanation supported by scientific studies to justify how fiber intake increases adiponectin levels. 

Thank you again for your valuable input.

Line 822 high-intensity interval training = (HIIT). Please abbreviate

We have abbreviated high-intensity interval training as HIIT: "High-intensity interval training (HIIT). Thank you

Reviewer 2 Report

Comments and Suggestions for Authors

This is a review about the potential of specific nutrients, diets, and physical activities to increase adiponectin levels and support of metabolic health. 

Overall, the article is well written and interesting to read.

However, more in depth reflections of the complex action of adiponectin are missing. The authors are recommended to focus more on the influence of nutrients and/or physical activity on adiponectin multimers/isoforms distribution patterns (PMID: 18820653, PMID: 16380500), and associated metabolic response types. Also the interplay between decreased levels of total adiponectin and the different adiponectin isoforms and NAFLD (PMID: 32715774,) is interesting in this context, and should be mentioned and discussed.

Furthermore, the dietary ω-6:ω-3 polyunsaturated fatty acid (PUFA) ratio and its association with adiposity and serum adiponectin levels is a matter of interest (PMID: 30092569) 

Comments on the Quality of English Language

minor editing warranted

Author Response

Dear Reviewer,

First of all, we would like to thank you for the valuable impulses that allowed us to improve the quality of the manuscript. All changes made are highlighted by yellow color, in the revised version of the manuscript, to facilitate the review process. Hoping that we have satisfied your requests as much as possible, we kindly ask you to re-evaluate our paper. 

The Authors

Reviewer 2

This is a review about the potential of specific nutrients, diets, and physical activities to increase adiponectin levels and support of metabolic health. 

Overall, the article is well written and interesting to read.

However, more in depth reflections of the complex action of adiponectin are missing. The authors are recommended to focus more on the influence of nutrients and/or physical activity on adiponectin multimers/isoforms distribution patterns (PMID: 18820653, PMID: 16380500), and associated metabolic response types. Also the interplay between decreased levels of total adiponectin and the different adiponectin isoforms and NAFLD (PMID: 32715774,) is interesting in this context, and should be mentioned and discussed. Furthermore, the dietary ω-6:ω-3 polyunsaturated fatty acid (PUFA) ratio and its association with adiposity and serum adiponectin levels is a matter of interest (PMID: 30092569) 

Thank you for your comments and suggestions.

We appreciate your feedback and have made the following revisions to address your concerns:

  1. Influence of Nutrients and/or Physical Activity on Adiponectin Multimers/Isoforms Distribution Patterns:
    • We have expanded section 4.7 to include the influence of nutrients and physical activities on the distribution patterns of adiponectin multimers/isoforms, citing studies that highlight their distinct biological activities and metabolic responses (Lara-Castro et al., 2006; Lara-Castro et al., 2008).
  2. Interplay Between Adiponectin Isoforms and NAFLD:
    • We have included a discussion on the interplay between decreased levels of total adiponectin and the different adiponectin isoforms in the context of NAFLD, referencing recent findings on this topic (Martínez-Huenchullán et al., 2023).
  3. Dietary ω-6:ω-3 Polyunsaturated Fatty Acid (PUFA) Ratio and Adiponectin Levels:
    • We have revised the text to discuss the dietary ω-6:ω-3 PUFA ratio and its association with adiposity and serum adiponectin levels, emphasizing the need for a balanced PUFA ratio for metabolic health (Torres-Castillo et al., 2018).

Thank you again for your valuable input.

Reviewer 3 Report

Comments and Suggestions for Authors

This review describes the role of adiponectin in maintaining lipid and glucose metabolism and preventing oxidative stress and inflammation in several tissues. The review first describes the function of different adipose tissues (about 6 pages) before describing adiponectin. This creates the expectation that the role of these different types of adipose in adiponectin function will be discussed later, which was not the case. Figure 2 is simplistic and lacks the required molecular pathways by which adiponectin induces its actions. The information in Table 2 seems repetitive of what is described in section 3.1.1. The information provided in section 3.1.1 is descriptive and lacks detailed molecular mechanisms and references. Overall, references are missing throughout the manuscript. After reading this extensive review, it is unclear to this reviewer how adiponectin regulates its many functions. It is mentioned that its structure is important for function. Still, it is unclear which form is the most active and if different forms activate the different receptors in different tissues. A model for the signaling pathways would be helpful.

Authors should consider removing the sections related to adipose tissue and adiponectin structure since these are not novel to focus mainly on the role of adiponectin.

Other comments:

1)        The description of adipogenesis in section 2.1 (lines 74-90) is very general. For the commitment and terminal differentiation, it is important to mention the specific transcription factors and molecules involved in the process.

2)        A period is missing in line 124.

3)        Remove the extra period in line 133.

4)        Capitalize “beige” in line 136

5)        Unclear what “the begging of the WAT” means in line 137

6)        Define Myf5 in line 145

7)        Define GPIHBP1 and FATP1 in lines 171 and 172

8)        Converted would be better than “mutated” in line 177

9)        Revise the sentence in lines 192-193; something is missing.

10)  The use of future tense in some sections should be revised. For example, in line 193. “The glycerol produced is used…”.

11)  Change “As regard” to “Regarding” In line 206.

12)  For section 2.3, cold-induced thermogenesis should be described before diet-induced thermogenesis for consistency.

13)  Correct norepinephrine in line 215 and the rest of the text

14)  Revise the sentence in line 251. MCP1 is not a kinase.

15)  Triglycerides are abbreviated in line 303 but not in previous sections where they were mentioned.

16)  Revise sentence in lines 318-319. Secretory vesicles are formed in the Golgi and then secreted at the plasma membrane.

17)  The sentences in line 333-334 is unclear.

18)  A reference is needed in the sentence in line 339 regarding ceramide and for the receptors in the rest of the paragraph.

19)  Information in lines 344-347 is a repetition of the information in lines 327-330.

20)  From the description of the structure of adiponectin it is unclear the differences between the globular forms and the timers and the MMW and HMW.

21)  Revise TNFa in line 599.

22)  Line 630, there is a period missing.

23)  For section 4.6, mixing data from human and mouse studies is confusing. It would be better to separate these studies in 2 paragraphs.

24)  Abbreviations should be revised throughout the manuscript. Some acronyms are not defined, while others are used and defined later in the manuscript.

Comments on the Quality of English Language

Revisions related to punctuation, abbreviations, and future tense in some sections need revision

Author Response

Dear Reviewer,

First of all, we would like to thank you for the valuable impulses that allowed us to improve the quality of the manuscript. All changes made are highlighted by yellow color, in the revised version of the manuscript, to facilitate the review process. Hoping that we have satisfied your requests as much as possible, we kindly ask you to re-evaluate our paper. 

The Authors

Reviewer 3

This review describes the role of adiponectin in maintaining lipid and glucose metabolism and preventing oxidative stress and inflammation in several tissues. The review first describes the function of different adipose tissues (about 6 pages) before describing adiponectin. This creates the expectation that the role of these different types of adipose in adiponectin function will be discussed later, which was not the case.

Thank you for your observation. We now added the missing information in order to better clarify the role of adiponectin related to adipose tissue. 

Figure 2 is simplistic and lacks the required molecular pathways by which adiponectin induces its actions. The information in Table 2 seems repetitive of what is described in section 3.1.1. 

Thank you for this comment. We detailed Figure 2. Table 2 reports the impact of the mediterranean diet on adiponectin levels whereas 3.1.1. reports the functions of adiponectin in different organs of the body. According to your suggestion we synthesized Table 1. 

The information provided in section 3.1.1 is descriptive and lacks detailed molecular mechanisms and references. 

Thank you for your comments. We added more details about the molecular mechanism mediated by adiponectin. Each paragraph in “3.1.1 Functions of adiponectin in different organs of the body” was detailed and the references were improved. 

Overall, references are missing throughout the manuscript. 

Thank you for your feedback. We have thoroughly revised the manuscript and added the necessary references throughout, addressing the concerns mentioned.

After reading this extensive review, it is unclear to this reviewer how adiponectin regulates its many functions. It is mentioned that its structure is important for function. Still, it is unclear which form is the most active and if different forms activate the different receptors in different tissues. A model for the signaling pathways would be helpful.

Thank you for your comment. We improved the manuscript underling that the larger oligomers, particularly HMW forms, are more effective in enhancing insulin sensitivity, exerting anti-inflammatory effects, and promoting fatty acid oxidation. However, also the globular form, even if it circulates in low abundance in human plasma may influence energy balance by promoting the oxidation of free fatty acids (FFA) in muscle mitochondria.

Authors should consider removing the sections related to adipose tissue and adiponectin structure since these are not novel to focus mainly on the role of adiponectin.

We appreciate the reviewer's feedback. In response, we have significantly reduced the sections related to adipose tissue and the structure of adiponectin to better focus on the role of adiponectin. Specifically, we have removed the following lines: 81-90, 113-116, 121, 124-129, 129-131, 190-194, 224-231, 250-252, and 279-287.

Other comments:

1)        The description of adipogenesis in section 2.1 (lines 74-90) is very general. For the commitment and terminal differentiation, it is important to mention the specific transcription factors and molecules involved in the process.

  1. We have now modified the paragraph on adipogenesis by adding the transcription factors and molecules involved.

2)        A period is missing in line 124. 

  1. We have now corrected the sentence.

3)        Remove the extra period in line 133. 

  1. The extra period was deleted.

4)        Capitalize “beige” in line 136.

  1. We have corrected the capital letter B.

5)        Unclear what “the begging of the WAT” means in line 137. 

  1. We have now changed the sentence.

6)        Define Myf5 in line 145. 

  1. We have now added the definition.

7)        Define GPIHBP1 and FATP1 in lines 171 and 172. 

  1. We have now added both definitions.

8)        Converted would be better than “mutated” in line 177. 

  1. We have now changed the verb.

9)        Revise the sentence in lines 192-193; something is missing. 

As suggested by the reviewer, some sentences from the first part have been eliminated, including lines 192-193.

10)  The use of future tense in some sections should be revised. For example, in line 193. “The glycerol produced is used…”. 

As suggested by the reviewer, some sentences from the first part have been eliminated, including line 193.

11)  Change “As regard” to “Regarding” In line 206. 

  1. We have now changed the word. 

12)  For section 2.3, cold-induced thermogenesis should be described before diet-induced thermogenesis for consistency. 

  1. We have now reversed the paragraphs of section 2.3.

13)  Correct norepinephrine in line 215 and the rest of the text.

  1. We have now changed the word. 

14)  Revise the sentence in line 251. MCP1 is not a kinase. 

  1. we added the correct term: chemokine. Part of line 251 has been deleted as suggested by the reviewer.

15)  Triglycerides are abbreviated in line 303 but not in previous sections where they were mentioned.

We have revised the manuscript to ensure that the abbreviation 'Triglycerides' (TG) is used consistently throughout the document. We have added the full definition 'Triglycerides (TG)' to the first occurrence and used the abbreviation 'TG' in subsequent sections.

16)  Revise sentences in lines 318-319. Secretory vesicles are formed in the Golgi and then secreted at the plasma membrane.

Ok. We have revised the sentence. 

17)  The sentences in line 333-334 are unclear.

  1. Thank you, we revised the sentences to better clarify the distribution of AdipoR1 and AdipoR2. 

18)  A reference is needed in the sentence in line 339 regarding ceramide and for the receptors in the rest of the paragraph.

Ok. Thank you. The reference is now added. (doi: 10.1038/nm.2277)

19)  Information in lines 344-347 is a repetition of the information in lines 327-330.

  1. Thank you. The information in lines 344-347 was removed. 

20)  From the description of the structure of adiponectin it is unclear the differences between the globular forms and the timers and the MMW and HMW.

  1. Thank you for your comment. We clarified better according to your previous comment. We clarified that the globular form is derived from the proteolytic cleavage of the full-length protein.

21)  Revise TNFa in line 599.

Thank you for your valuable feedback. "TNFa" has been revised to "TNF-α".

22)  Line 630, there is a period missing.

Thanks

23)  For section 4.6, mixing data from human and mouse studies is confusing. It would be better to separate these studies in 2 paragraphs.

Thank you for your valuable feedback. We have revised Section 4.6 to clearly separate the data from human and preclinical studies. This has been achieved by organizing the section into distinct subsections. Additionally, we have divided the original table into two separate tables to further clarify the distinction between preclinical and human studies. This structure ensures a clearer presentation of the data and addresses the concern of mixing human and mouse study results. We appreciate your suggestions and believe this change enhances the clarity and readability of our manuscript.

24)  Abbreviations should be revised throughout the manuscript. Some acronyms are not defined, while others are used and defined later in the manuscript.

Thank you. We have carefully examined all the abbreviations used in the manuscript and have made the following corrections to ensure that they are all defined at their first occurrence and used consistently throughout the text: We added the definition of each acronym at its first occurrence in the text. We created a specific section for acronyms used in the manuscript and placed it before the list of references for easy reference. We verified that all abbreviations are consistent throughout the document, including tables and figure captions. In addition, we have added a list of acronyms used in the manuscript before the references section to help the reader easily understand the abbreviations.

25) Comments on the Quality of English Language. Revisions related to punctuation, abbreviations, and future tense in some sections need revision.

We revised and corrected the punctuation, abbreviations and future tense in the manuscript. We hope that these changes adequately address your comments and improve the clarity and coherence of our manuscript.

Round 2

Reviewer 1 Report

Comments and Suggestions for Authors

The authors have satisfactorily addressed most of my concerns. In particular, the authors have greatly improved the fluency and readability of the manuscript. 

Author Response

Thank you very much

Reviewer 3 Report

Comments and Suggestions for Authors

Most of my previous comments were addressed by the authors. However, Figure 2 was not revised as suggested: “Figure 2 is simplistic and lacks the required molecular pathways by which adiponectin induces its actions.”

A new figure summarizing the molecular mechanisms in the different tissues is needed, maybe in the discussion section. This would be important for readers to better understand the complexity of adiponectin actions.

Author Response

Dear Auditor,

Thank you for your constructive comments. We have included the new Figure 2 in section 3.1.1, which summarises the molecular mechanisms in different tissues. We hope that this revision meets your expectations and improves readers' understanding of adiponectin's actions.

We remain available for further suggestions.

Best regards,

the authors